# SERA: Soft Ensemble Reliability Aggregation for Robust Multi-Agent Reinforcement Learning

## Abstract

Bootstrapped temporal-difference learning inherently introduces variance into value estimates, which often destabilizes learning due to value function oscillation between over- and under-estimation. Overestimation is commonly mitigated through pessimistic critic updates, but such bias-based approaches can introduce underestimation and do not address the estimation variance, which is often amplified in multi-agent reinforcement learning (MARL) due to its inherent learning complexities. To address this, we propose SERA, a soft ensemble reliability aggregation framework designed to reduce value estimation variance through reliability-aware critic aggregation. SERA constructs targets through soft reliability-weighted aggregation of critic estimates and introduces a novel decorrelation mechanism that adaptively tunes each critic's learning rate based on temporal-difference error uncertainty and the variance of target estimation error. This leads to more stable and reliable target estimation during training. Experiments on a wide range of multi-agent continuous-control benchmarks from MuJoCo and PettingZoo show that SERA consistently outperforms strong twin-critic and ensemble baselines, achieving performance improvements of up to 41.1%. We further demonstrate that the same framework generalizes well to single-agent continuous-control tasks, providing gains of up to 31.25% over established methods.

## 1 Introduction

Deep reinforcement learning (DRL) builds upon classical value-based methods and the actor–critic paradigm, both grounded in the principle of Temporal-Difference (TD) learning (Sutton et al., 1998). Even in the tabular setting, where convergence is guaranteed, TD-based Q-learning is known to produce overestimated value estimates during training despite asymptotic convergence (Tsitsiklis, 1994). This overestimation arises from uncertainty in value estimates and the effect of Jensen's inequality in the maximization step (Thrun & Schwartz, 2014), and is further exacerbated under function approximation (Kim et al., 2019; Duan et al., 2020).

This issue continues to persist under function approximation with deep neural networks such as Deep Q-Networks (DQN) (Mnih et al., 2015), deep deterministic policy gradient (DDPG) (Lillicrap et al., 2015), etc. In discrete-action settings,several methods have been proposed to mitigate overestimation such as Double DQN (Van Hasselt et al., 2016), Bootstrapped DQN (Osband et al., 2016), and ensemble-based approaches such as MeanQ (Liang et al., 2022), and Maxmin Q-learning (Lan et al., 2020). In continuous control, actor–critic methods such as DDPG (Lillicrap et al., 2015) suffer from similar overestimation issues under function approximation. To address this, algorithms such as Twin Delayed Deep Deterministic Policy Gradient (TD3) (Fujimoto et al., 2018) and Soft Actor–Critic (SAC) (Haarnoja et al., 2018) employ twin critics. Although these conservative critic updates can suppress overestimation, they may also push value estimates downward and still leave fluctuations in the target estimates insufficiently controlled.

In addition to bias-related issues, variance in bootstrapped value targets remain a major cause of instability in temporal-difference learning. Although TD learning is generally preferred over Monte Carlo methods for its lower variance, reliance on bootstrapped targets introduces an additional source of noise, as updates are computed from randomly sampled transitions rather than exact expectations (Xu et al., 2020). Moreover,

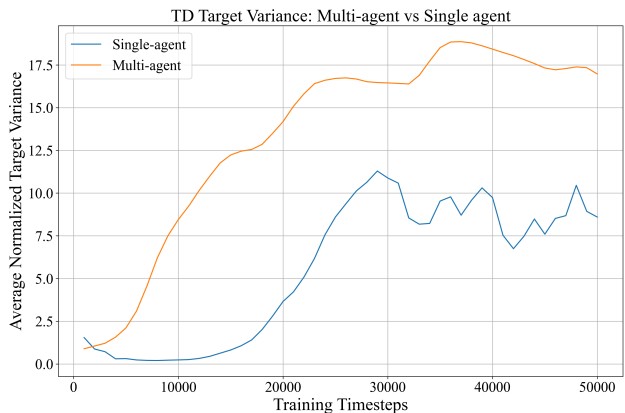

Figure 1: TD target variance during training for single-agent and multi-agent RL settings. The multi-agent setting exhibits consistently higher normalized target variance over a batch during training, indicating increased instability in value estimation.

in the presence of stochastic rewards and non-zero learning rates, this injected variance can propagate through successive updates, resulting in unstable value estimates (Pan & Schölkopf, 2025). When using function approximators such as neural networks, the bootstrapped value estimates are further affected by the bias–variance trade-off of the approximator, causing oscillations between under- and overestimation (Bengio et al., 2020). Thus, deep reinforcement learning is affected by variance-related issues from two sources: one arising from TD learning itself and the other from the inherent behavior of neural networks. As a result, reducing target estimation variance remains one of the challenges to address for efficient learning.

Multi-agent reinforcement learning (MARL) introduces additional sources of instability beyond those encountered in single-agent settings, including non-stationarity (Li et al., 2021), credit assignment difficulties (Sunehag et al., 2017), and partial observability (Omidshafiei et al., 2017). These factors increase the variability of value estimates and make critic learning significantly more difficult. As illustrated in Fig. 1, TD-target variance in multi-agent environments remains consistently higher than in comparable single-agent settings throughout training. Following Lyu et al. (2021), this additional variability can be attributed to two major components: Multi-Action Variance (MAV), caused by the stochastic behavior of other agents, and Multi-Observation Variance (MOV), arising from differences in their local observations and histories. Both effects contribute to noisier TD targets and less stable policy updates, and their impact becomes increasingly severe as the number of interacting agents grows. Consequently, centralized critics in MARL are considerably more prone to unstable value estimation. Despite the growing use of ensemble critics in reinforcement learning, most existing approaches largely inherit design choices from single-agent methods and do not explicitly address estimation variance as a primary source of training instability. This raises an important question: can directly controlling estimation variance improve the stability and reliability of multi-agent learning?

To address this limitation we propose a soft ensemble reliability aggregation (SERA) framework for multi-agent reinforcement learning. Unlike prior ensemble methods that (i) rely on uniform averaging (e.g., MeanQ (Liang et al., 2022) in single-agent discrete settings), (ii) primarily encourage exploration (e.g., SUNRISE (Lee et al., 2021)), or (iii) reduce bias via pessimistic minimization of Q-values (e.g., MATD3 and MASAC in MARL), SERA treats estimation variance as a first-class learning signal that jointly governs target construction and critic updates. In particular, critic outputs are adaptively weighted based on their estimated reliability through a median-guided soft reliability aggregation, producing low-variance and robust training targets that attenuate noisy or inconsistent predictions while emphasizing more stable critic estimates. To further reduce correlation among ensemble members, we introduce a novel variance-aware decorrelation framework that adaptively adjusts each model critic's learning rate using temporal-difference error uncertainty and the variance of target estimation error. Together, these components improve target estimation stability, reduce critic correlation, and enable more sample-efficient learning. We evaluate SERA in

both multi-agent and single agent reinforcement learning settings to examine its effectiveness and generality across different learning scenarios. The key contributions are summarized as follows:

- *Reliability-Aware Target Aggregation:* We develop an ensemble critic framework in which the target values are constructed by assigning sample-wise soft reliability weights based on each critic's deviation from the median critic estimate, producing a unified and more stable target estimate.

- *Decorrelation Mechanism:* We propose a novel adaptive *learning rate* based critic decorrelation mechanism that adapts based on both temporal-difference error uncertainty and target estimation error variance.

- *Benchmarking in Multi-Agent Systems:* We adapt SERA to the centralized training–decentralized execution (CTDE) setting and evaluate it on five multi-agent continuous control benchmarks from MuJoCo and PettingZoo (Terry et al., 2021). Under a fixed interaction budget of 300K steps, SERA achieves performance improvements of up to 41.1% and an average improvement of 28.7% compared to strong baselines.

- *Generalization to Single-Agent Continuous-Control Tasks:* We further extend the framework to single-agent continuous control, where SERA consistently surpasses recent ensemble-based methods and achieves gains of up to 31.25%.

## 2 RELATED WORKS

In this section, we briefly discuss the works that are closely related to our study.

### 2.1 Centralized Training with Decentralized Execution (CTDE)

Centralized Training with Decentralized Execution (CTDE) is the standard setup for MARL, enabling agents to leverage global information during training while maintaining decentralized policies at test time. Within CTDE, off-policy actor–critic baselines include MADDPG (Lowe et al., 2017), MAAC (Iqbal & Sha, 2019), and MATD3 (Ackermann et al., 2019), while entropy-regularized variants such as MASAC improve robustness. On-policy methods such as MAPPO (Yu et al., 2022) offer improved stability but at the cost of sample efficiency. Despite these advances, most CTDE methods continue to rely on standard target construction and critic update rules, leaving the variance of centralized value estimates largely uncontrolled in challenging multi-agent environments.

### 2.2 Ensemble-based Approaches

A number of ensemble-based methods have been proposed for multi-agent continuous control under the CTDE setting. EMAX (Lukas Schäfer & Mguni), and Implicit Ensemble Training (IET) (Shen & How, 2023) leverage ensembles primarily to encourage exploration, representation diversity, or robustness, rather than to explicitly control instability arising from noisy value estimates during critic learning. As a result, estimation variance remains largely untreated as a first-class factor in stabilizing critic updates.

In single-agent discrete-action learning, MeanQ (Liang et al., 2022) reduces overestimation by lowering target variance via ensemble averaging. However, it is formulated for discrete action-space and does not address variance propagation or instability in bootstrapped actor–critic learning, leaving estimation variance largely unregulated in continuous-control MARL. Methods such as ACE (Zhang & Yao, 2019) improve continuous-control learning by using an ensemble of actors to search for actions with higher critic values. This helps the policy update explore a richer set of candidate actions. However, ACE mainly introduces diversity on the actor side and does not explicitly consider how reliable each critic estimate is when forming target values. In contrast, SERA operates on the critic ensemble directly, assigning sample-wise reliability weights to target critics according to their deviation from a robust median anchor. Similarly, SUNRISE (Lee et al., 2021) leverages ensemble diversity to enhance exploration and robustness, rather than directly addressing variability in value estimation during critic updates.

Existing ensemble-based reinforcement learning methods typically improve learning through three main strategies: averaging critic estimates, using minimum operators for conservative target selection, or leveraging ensemble diversity to encourage exploration. In most existing methods, these ideas are handled separately. Averaging-based approaches primarily aim to reduce fluctuations in value estimates, while minimum-based targets are mainly used to control overestimation. Methods that exploit ensemble disagreement typically use it for exploration purposes instead of directly stabilizing the critic learning process.

SERA instead treats estimation variance as an important quantity to manage during training. The framework combines reliability-based target aggregation with adaptive critic updates so that unreliable critic estimates contribute less to learning, while diversity within the ensemble is still maintained. As a result, variance is controlled both when forming the bootstrapped TD targets and when updating the critic networks. This is different from most existing ensemble actor–critic methods, which usually emphasize either target aggregation or exploration independently.

## 3 Preliminaries

### 3.1 Markov Games

We model the multi-agent environment as a *Markov game* (Littman, 1994), which extends the standard Markov Decision Process (MDP) framework to settings involving multiple interacting agents. The game is represented by the tuple $(\mathcal{I}, \mathcal{S}, \{\mathcal{A}_i\}_{i \in \mathcal{I}}, \mathcal{P}, \{R_i\}_{i \in \mathcal{I}}, \{\Omega_i\}_{i \in \mathcal{I}}, \gamma)$, where $\mathcal{I}$ denotes the collection of agents and $\mathcal{S}$ corresponds to the global state space. At each time step, the agents jointly execute an action vector $\mathbf{a} = (a_1, \ldots, a_N) \in \mathcal{A}$, resulting in a state transition governed by the probability mapping $\mathcal{P} : \mathcal{S} \times \mathcal{A} \to \text{Dist}(\mathcal{S})$. Each agent $i$ observes a local observation $o_i \in \Omega_i$ and receives a reward signal determined by the reward function $R_i(s, \mathbf{a})$. Decision making is performed through stochastic policies of the form $\pi_i(a_i \mid o_i)$. The objective of every agent is to maximize the expected discounted cumulative reward given by $J_i = \mathbb{E}\left[\sum_{t=0}^{T} \gamma^t r_i^{(t)}\right]$, where $\gamma \in (0, 1]$ represents the discount factor. In this work, we specifically consider cooperative multi-agent settings in which all agents aim to achieve a common or mutually aligned objective.

### 3.2 Centralized Training and Decentralized Execution (CTDE)

CTDE paradigm decouples training from execution. During training, critics are allowed to access global information, while during execution each agent acts using only its local observations. Let the global state space be denoted as $\mathcal{S}$, the action space of agent $i$ is denoted as $A_i$, and $O_i$ its local observation space. Each agent follows a stochastic policy $\pi_i : O_i \to \text{Dist}(A_i), \quad a_i \sim \pi_i(\cdot \mid o_i)$, which depends only on its local observation $o_i \in O_i$. The critic is trained with centralized information. For agent $i$, the action-value function is defined as

$$Q_i^\pi(s, a_1, \ldots, a_N) = \mathbb{E}\left[\sum_{t=0}^{\infty} \gamma^t r_i^t \,\bigg|\, s_0 = s, \; a_j \sim \pi_j(o_j) \; \forall j\right],$$

where $s \in S$ is the global state, $(a_1, \ldots, a_N)$ denotes the joint action, and $\gamma \in (0, 1)$ is the discount factor.

Each agent seeks to maximize its expected return, given by $J_i(\pi_i) = \mathbb{E}_{s \sim \rho^\pi, a \sim \pi}\left[Q_i^\pi(s, a_1, \ldots, a_N)\right]$. While the critics benefit from the global state to mitigate non-stationarity, the policies $\pi_i$ only require local observations $o_i$, making the approach suitable for decentralized deployment under partial observability. For more details, the readers are referred to Lowe et al. (2017).

### 3.3 Temporal Difference Learning and Value Bootstrapping

Critics are commonly trained using temporal-difference (TD) learning. For a critic $Q_i$ with parameters $\theta_i$, the update is based on a bootstrapped target of the form

$$y_i = r_i + \gamma Q_i(s', \mathbf{a}'; \theta_i^-), \tag{1}$$

where $\bar{\theta}_i$ denotes a target network that is updated more slowly than the online parameters. This delayed update helps maintain stability by limiting rapid changes in the regression target.

Temporal-difference learning estimates future returns using its own current value predictions. Because of this recursive update structure, estimation errors can accumulate over time and influence later updates. Under function approximation, the bootstrapped targets may therefore contain both bias and variance from inaccurate value estimates, which can make training unstable (Pan & Schölkopf, 2025; Bengio et al., 2020).

These effects are further amplified in multi-agent environments. The joint action $\mathbf{a}' = (a'_1, \ldots, a'_N)$ is generated by multiple agents whose policies evolve concurrently, making the target distribution non-stationary. As a result, TD targets can become increasingly noisy during training. To address this issue, we introduce SERA, which stabilizes target estimation through reliability-weighted aggregation of critic outputs.

### 3.4 Ensemble Critics

In reinforcement learning, an ensemble refers to the use of multiple value functions trained for the same task. The collective behavior of an ensemble can be used to quantify uncertainty or to produce a lower-variance estimate, compared to a single estimator. This is used by prior works like Lee et al. (2021); Liang et al. (2022); Fujimoto et al. (2018).

Let $\{Q_{\theta_k}\}_{k=1}^K$ denote a collection of $K$ critics, each with its own parameters. For a given state-action pair $(s, \mathbf{a})$, the ensemble yields a set of predictions $\{Q_{\theta_k}(s, \mathbf{a})\}_{k=1}^K$. Maintaining multiple critics can help stabilize learning by mitigating noisy estimates and providing a measure of uncertainty through their disagreement. In this work, we make use of this ensemble to form more reliable targets during training.

## 4 SERA: Soft Ensemble Reliable Aggregation

In this section, we present the main components of the proposed Soft Ensemble Reliable Aggregation (SERA) framework. While SERA builds on the general idea of using multiple critics to stabilize temporal-difference learning, it takes a different direction by explicitly estimating how reliable each critic is and using this information to reduce noise in target estimation. This notion of reliability is used consistently in both how targets are formed and how critics are trained.

At the core of SERA is a soft aggregation scheme (reliability-aware target aggregation), where critics that behave more consistently have a stronger influence on the target, while those that deviate significantly are naturally down-weighted. Alongside this, we introduce a novel decorrelation mechanism designed to keep the critics sufficiently diverse, avoiding redundancy and improving overall stability. We begin by describing the reliability-aware target aggregation, and then present the proposed decorrelation strategy.

### 4.1 Reliability-aware Target Aggregation

To obtain a more stable target, we use a sample-wise aggregation scheme that combines the robustness of the median with an adaptive notion of reliability based on per-sample agreement among critics.

Let $\{Q_{\theta_i}(s, a)\}_{i=1}^N$ denote the ensemble of critics, and let $\{Q_{\bar{\theta}_i}(s, a)\}_{i=1}^N$ represent the corresponding target critics. For each sample $b$ in a mini-batch, we first compute the median anchor using the target critics:

$$Q_{\mathrm{med}}^{(b)}(s, a) = \mathrm{median}\left(\{Q_{\bar{\theta}_i}^{(b)}(s, a)\}_{i=1}^N\right), \tag{2}$$

and denote its index by $m_b$. This median serves as a robust reference point and reduces the effect of extreme critic estimates.

We then measure how much each critic deviates from this reference:

$$d_i^{(b)} = \left| Q_{\bar{\theta}_i}^{(b)}(s, a) - Q_{\mathrm{med}}^{(b)}(s, a) \right|. \qquad \forall i \in \{1, \ldots, N\} \tag{3}$$

Based on these deviations, we assign weights to the non-median critics using a soft exponential scheme:

$$w_i^{(b)} = \frac{\exp\left(-d_i^{(b)}/\tau\right)}{\sum_{k \neq m_b} \exp\left(-d_k^{(b)}/\tau\right)}, \qquad \tau > 0, \tag{4}$$

where $\tau$ controls how sharply the weights concentrate. Since the median critic has zero deviation by definition, it is excluded from the weighting procedure to prevent it from dominating the aggregation. This design assigns larger weights to critics whose predictions stay nearer to the median estimate, which is the central consensus. Remark 1 formally establishes that critics with smaller deviations from the consensus always obtain higher weights.

**Remark 1** (Sample-wise reliability ordering of SERA). *For any two non-median critics $i, j$ such that $i, j \neq m_b$, $d_i^{(b)} < d_j^{(b)} \Rightarrow w_i^{(b)} > w_j^{(b)}$. Also, if for all $b \in \mathcal{B}$, $d_i^{(b)} \leq d_j^{(b)}$, with strict inequality for at least one sample, then $\frac{1}{|\mathcal{B}|} \sum_{b \in \mathcal{B}} w_i^{(b)} > \frac{1}{|\mathcal{B}|} \sum_{b \in \mathcal{B}} w_j^{(b)}, \forall i, j \neq m_b$*

Therefore, critics that remain closer to the consensus throughout the mini-batch are assigned larger average reliability weights under SERA. This weighting behavior shapes the final aggregation by reducing the influence of outlier critics. The resulting aggregated estimate from the non-median critics is:

$$Q_{\text{sera}\backslash\text{med}}^{(b)}(s, a) = \sum_{i \neq m_b} w_i^{(b)} Q_{\bar{\theta}_i}^{(b)}(s, a). \tag{5}$$

We then combine this estimate with the median anchor to form the final SERA target:

$$Q_{\text{SERA}}^{(b)}(s, a) = \alpha \, Q_{\text{med}}^{(b)}(s, a) + (1 - \alpha) \, Q_{\text{sera}\backslash\text{med}}^{(b)}(s, a), \tag{6}$$

where $\alpha \in (0, 1)$ controls the trade-off between robustness and adaptivity.

**Remark 2** (Median-anchored reliability estimation). *Since the true target $y^\star$ is not directly accessible, the median is used as a robust anchor for comparing critic estimates. Its resistance to outliers makes it a robust reference for measuring deviations (Hampel, 1971; Rousseeuw & Leroy, 2003).*

In summary, the aggregation is performed at the individual sample level, assigning greater importance to critics that align more closely with the central tendency and suppressing the influence of outlier estimates. We now analyze some fundamental properties of the method, particularly the critic ranking induced by the weights and the characteristics of the final target.

**Proposition 4.1** (Convexity and Boundedness). *For each sample $b$, $Q_{\text{SERA}}^{(b)}$ is a convex combination of the target critic estimates. Consequently, the SERA estimate is bounded by the ensemble's extremes:*

$$\min_i Q_{\bar{\theta}_i}^{(b)} \leq Q_{\text{SERA}}^{(b)} \leq \max_i Q_{\bar{\theta}_i}^{(b)}.$$

Thus, unlike min-based aggregation such as MATD3 and MASAC, $Q_{\text{SERA}}^{(b)}$ does not reduce to the minimum critic estimate, thus avoiding excessive pessimistic underestimation. At the same time, it does not exceed the largest estimate, ensuring that the aggregation remains within the range of the ensemble predictions. This establishes bounded avoidance of extreme critic aggregation.

**Proposition 4.2** (Variance reduction relative to a single critic). *Suppose the target critic estimates satisfy $Q_{\bar{\theta}_i}^{(b)} = y^\star + \varepsilon_i$, where $\mathbb{E}[\varepsilon_i] = 0$ and $\text{Var}(\varepsilon_i) = \sigma^2$. Let $\sum_{i=1}^{N} \text{Cov}(\tilde{w}_i, \varepsilon_i^2) < 0$, where $\tilde{w}_i^{(b)} = \begin{cases} \alpha, & i = m_b, \\ (1 - \alpha) w_i^{(b)}, & i \neq m_b. \end{cases}$ Then, the SERA target satisfies*

$$\text{Var}\left(Q_{\text{SERA}}^{(b)}\right) < \sigma^2 = \text{Var}\left(Q_{\bar{\theta}_i}^{(b)}\right), \forall i \in \{1, \ldots, N\}$$

.

Thus, under the stated assumptions, SERA achieves a lower target variance than any individual critic whenever $\sum_{i=1}^{N} \text{Cov}(\tilde{w}_i, \varepsilon_i^2) < 0$. This condition reflects the desired behavior of the reliability-aware weighting scheme: critics with larger estimation errors tend to receive smaller weights, reducing their influence on the aggregated target and leading to a more stable estimate.

**Comparison with mean aggregation:** Proposition 4.2 shows that SERA produces a lower-variance target than an individual critic whenever the covariance condition stated in the proposition is satisfied. When all critic errors are independent and identically distributed, uniform averaging is the variance-optimal estimator, with variance $\sigma^2/N$, as established by classical estimation theory. In practice, however, ensemble reinforcement learning typically seeks to reduce error correlation by encouraging diversity among critics. This is commonly achieved through different network initializations, heterogeneous critic architectures, and different optimization trajectories, which naturally lead to critics with different levels of estimation reliability. Under these conditions, SERA assigns smaller weights to critics whose estimates deviate more from the ensemble median, thereby limiting the influence of less reliable estimates while retaining the advantages of ensemble aggregation. Although this does not imply that SERA is universally superior to uniform averaging, the sufficient-condition analysis together with the synthetic experiments indicates that reliability-aware aggregation can achieve lower empirical variance heterogeneous ensembles where critic reliability differs (see Appendix B).

Proposition 4.1 and 4.2 together show that the SERA target remains within the range of the ensemble predictions while reducing the variance of the aggregated estimate. As a result, critics that produce unstable or highly inconsistent estimates have a smaller influence on the final target, without forcing the update toward overly pessimistic values as in hard minimum selection (MATD3 or MASAC). This makes the TD targets less affected by noisy or outlier predictions, leading to more stable critic learning dynamics during training.

Maintaining diversity among ensemble critics is important because shared replay data and a common optimization objective can gradually lead to correlated estimation errors, reducing the benefits of ensemble aggregation. To mitigate this effect, SERA promotes critic diversity through independent parameter initialization, heterogeneous network architectures, and the variance-adaptive learning rate strategy described in Section 4.2. Together, these mechanisms encourage decorrelated critic behavior and improve the robustness of the reliability-aware aggregation.

## 4.2 Decorrelation Strategy

Effective variance reduction through ensemble aggregation relies on maintaining diversity among critics, as averaging identical estimates provides no benefit. Accordingly, SERA encourages decorrelation so that critics retain sufficiently diverse estimation errors, enabling meaningful variance reduction. To promote diversity among the ensemble critics, three complementary strategies are employed. First, all networks are initialized with different random parameters so that each critic begins training from a unique starting point. Second, the critics are designed with heterogeneous hidden-layer architectures, encouraging them to capture different feature representations during learning. In addition, a learning-rate-driven decorrelation strategy is introduced to further reduce similarity among the critic updates.

### 4.2.1 Variance-Adaptive Learning Rate

In addition to the reliability-aware target construction, we incorporate a novel mechanism that adjusts each critic's update strength according to the spread of the ensemble.

Let $Q^\star(s, a)$ denote the true action–value for a given state–action pair $(s, a)$. $\{Q_{\theta_i}(s, a)\}_{i=1}^N$ denote the ensemble of critics, and $\{Q_{\bar{\theta}_i}(s, a)\}_{i=1}^N$ the corresponding target critics. For a transition $(s, a, r, s')$, the temporal-difference (TD) target is defined as: $y = r + \gamma Q_{\bar{\theta}}(s, a)$.

To characterize uncertainty in learning, we introduce two variance terms. The uncertainty associated with the critic's current estimate is defined as

$$P_t = \mathrm{Var}\big(Q_t - Q^\star(s, a)\big) \tag{7}$$

and is termed as estimation variance, while the uncertainty of the TD target is defined as

$$R_t = \mathrm{Var}\big(y - Q^\star(s, a)\big), \tag{8}$$

and is termed as measurement variance. The estimation variance reflects the uncertainty in the current critic estimate, while the measurement variance captures the noise present in the TD target. The following lemma, termed Variance-aware adaptive critic step size, can be established.

**Lemma 4.3** (Variance-aware adaptive critic step size). *Assume that the critic estimation error $e_t = Q_t - Q^\star(s, a)$ and the TD target error $\varepsilon_t = y_t - Q^\star(s, a)$ are zero-mean, finite variance and uncorrelated. Then, the step size $\beta_t$ that minimizes the expected mean-square error $\mathbb{E}(Q_{t+1} - Q^\star(s, a))^2$ is given by*

$$\beta_t^\star = \frac{P_t}{P_t + R_t}.$$

**Remark 3** (Practical validity of assumptions). *Lemma 4.3 derives the variance-aware critic step size under the assumptions of zero-mean and uncorrelated estimation errors for optimal conditions. These assumptions are introduced only to obtain a tractable analytical result and are not expected to hold exactly in deep reinforcement learning, where bootstrapped TD targets are generally biased. SERA does not require these conditions to hold exactly in practice; the resulting step-size is used as an uncertainty-gated heuristic, akin to adaptive optimization and filtering methods in non-linear systems. Empirically, this variance-adaptive update remains effective in MARL.*

The adaptive learning rate reflects a balance between the trustworthiness of the current critic's prediction estimate $P$ and the reliability of incoming TD targets $R$. This balance ensures stable learning while still allowing rapid adaptation when reliable information is available. The sensitivity analysis of $\beta_t^\star$ is provided in the appendix. This form is inspired by the Kalman gain, which balances the uncertainty in the estimate and the target (Li et al., 2015).

### 4.2.2 Practical design of the adaptive learning rate

In practice, the true variances $P_t$ and $R_t$ are not directly observable, as they depend on the unknown quantity $Q^\star(s, a)$. We therefore approximate them using empirical batch statistics. For each critic $i$, we estimate the measurement variance as

$$R_{\text{gain},i} = \text{Var}_{j \in \mathcal{B}} \left( Q_{\bar{\theta}_i}(s_j', \mathbf{a}_j') - Q_{\text{SERA}}(s_j', \mathbf{a}_j') \right), \tag{9}$$

where $\mathcal{B}$ denotes the mini-batch. Here, $Q_{\text{SERA},j}$ serves as the aggregated target, and the deviation from it reflects the critic's estimation error.

Within the ensemble framework, every critic is updated using its own adaptive learning rate, allowing the update magnitude to vary according to both the critic-specific uncertainty and the disagreement from the ensemble consensus. For the $i$-th critic, the prediction variance at training step $t$, denoted by $v_{Q,i}^{(t)}$, is computed from the variance of the temporal-difference (TD) residuals over the sampled minibatch. This quantity is tracked using an exponential moving average to obtain a stable estimate during training:

$$v_{Q,i}^{(t)} \leftarrow \alpha_Q v_{Q,i}^{(t-1)} + (1 - \alpha_Q) \text{Var}_{j \in \mathcal{B}} \left( \delta_j^{(i)} \right), \tag{10}$$

where $\delta_j^{(i)} = y_j - Q_{\theta_i}(s_j, \mathbf{a}_j)$ is the TD error of critic $i$ and $\alpha_Q$ controls the smoothness.

The estimation variance $P$ is then updated by accumulating the prediction-variance:

$$P_t^{(i)} \leftarrow \alpha_P P_{t-1}^{(i)} + (1 - \alpha_P) v_{Q,i}^{(t)}. \tag{11}$$

Here, $P_{t-1}^{(i)}$ denotes the posterior estimation variance of critic $i$ after incorporating all information available up to time $t - 1$. This term captures the remaining uncertainty in the critic's value estimate.

The adaptive scaling factor for critic $i$, from Lemma 4.3, is defined as

$$\kappa_t^{(i)} = \frac{P_t^{(i)}}{P_t^{(i)} + R_{\text{gain},i} + \eta}, \qquad \kappa_t^{(i)} \in (0, 1). \tag{12}$$

where $\eta > 0$ ensures numerical stability. The critic parameters are then updated with a variance-adaptive gradient step:

$$\theta^{(i)} \;\leftarrow\; \theta^{(i)} + \frac{\alpha_Q}{S} \sum_{j=1}^{S} \kappa_t^{(i)} \, \delta_j^{(i)} \, \nabla_{\theta^{(i)}} Q_{\theta^{(i)}}(s_j, \mathbf{a}_j), \tag{13}$$

This design is inspired by the Kalman filter under simplifying assumptions; further details are provided in the appendix. To understand the relationship between $R$ and $R_{\text{gain}}$ is shown in appendix.

### 4.3 SERA Algorithm and its Complexity

SERA combines the novel learning rate based decorrelating mechanism along with the reliability-aware target formulation. The whole training procedure is shown in Algorithm 1 (please refer the supplementary file). We chose MASAC updating rules for our base.

**Computational Complexity:** The per-iteration computational cost of the proposed SERA framework can be expressed as $\mathcal{O}(NS\,C_Q + MS\,C_\pi)$, where the individual terms represent different stages of the training procedure. In this formulation, $N$ corresponds to the number of critic networks in the ensemble, $S$ denotes the minibatch size, and $M$ indicates the number of decentralized policy networks. The terms $C_Q$ and $C_\pi$ represent the computational costs associated with the forward and backward passes of a single critic and actor network, respectively. The component $NS\,C_Q$ arises from computing and updating all ensemble critics, while $MS\,C_\pi$ captures the cost of updating the actor policies. In addition, the aggregation operation grows linearly with respect to both the ensemble size and minibatch size, introducing only a minor overhead compared to the overall neural network optimization process.

## 5 Experimental Evaluations

We evaluate the proposed SERA framework on a diverse collection of cooperative multi-agent continuous control tasks from the MuJoCo and PettingZoo benchmarks. To examine the scalability of the method under varying levels of coordination complexity, the selected environments include systems ranging from two to six agents. SERA is compared against four representative CTDE-based baselines, namely IET, MATD3, MASAC, and MAPPO, using identical network architectures and training protocols for fair comparison. Additional implementation details and hyperparameter configurations are included in the appendix. Performance is measured using the average episodic return throughout training, and results are reported with standard deviation across 10 random seeds to evaluate both stability and consistency.

Alongside SERA, we also consider **Ensemble Mean** as a comparison baseline to examine the effect of variance-aware target aggregation. Experiments are conducted on five multi-agent continuous control benchmarks, including three MaMuJoCo environments and two cooperative PettingZoo tasks. These benchmarks cover a range of coordination challenges, interaction patterns, and control difficulties. We focus on continuous-action environments since value estimation variance can become more pronounced in continuous domains because of the larger action space and the coupled learning dynamics among agents. In addition, MASAC and similar actor–critic approaches are mainly developed for continuous control, making these benchmarks appropriate for evaluating the proposed framework.

Table 1: Performance comparison of SERA with baseline algorithms across multi-agent environments.

| Environment | SERA | Ensemble Mean | IET | MASAC | MATD3 | MAPPO |
|---|---|---|---|---|---|---|
| HalfCheetah (6 agents) | **5520** $\pm$ 711 | $4831 \pm 721$ | $4608 \pm 783$ | $4500 \pm 802$ | $3815 \pm 912$ | $3623 \pm 812$ |
| Pusher (3 agents) | **-27** $\pm$ 8 | $-30 \pm 7$ | $-29 \pm 9$ | $-31 \pm 8$ | $-50 \pm 11$ | $-57 \pm 13$ |
| Hopper (3 agents) | **2380** $\pm$ 519 | $2190 \pm 611$ | $1701 \pm 600$ | $1530 \pm 200$ | $840 \pm 400$ | $790 \pm 100$ |
| Multiwalker (3 agents) | **-27.4** $\pm$ 5 | $-39 \pm 5$ | $-46.5 \pm 5$ | $-51 \pm 10$ | $-56 \pm 10$ | $-60.3 \pm 10$ |
| Simple Spread (3 agents) | **-9** $\pm$ 2 | $-13 \pm 2.3$ | $-14 \pm 4$ | $-16 \pm 5.1$ | $-17.5 \pm 6$ | $-18 \pm 2$ |

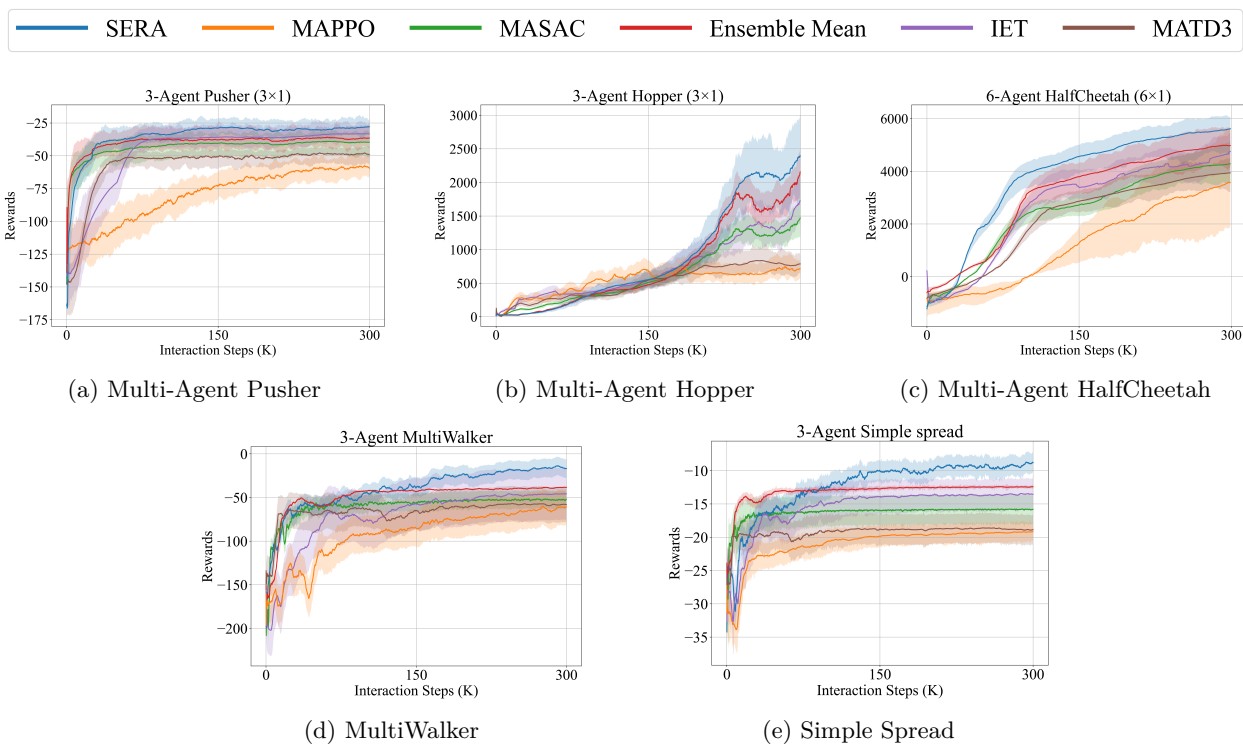

Figure 2: Performance comparison across different multi-agent cooperative environments.

## 5.1 Results

Fig. 2 presents the average reward curves and standard deviation profiles for both the MuJoCo and PettingZoo environments. The rewards are averaged over ten independent runs to reflect not only overall performance but also the consistency of training behavior across different random seeds. The selected benchmarks include a variety of coordination patterns and control dynamics, allowing evaluation under diverse multi-agent settings.

The results in Fig. 2 show that SERA consistently achieves stronger performance than all compared baselines across the tested environments. It also performs better than its ensemble counterpart, **Ensemble Mean**. Although simple ensemble averaging can sometimes improve over standard baselines, its gains are not consistently maintained against IET. SERA, on the other hand, shows stable improvements across tasks, highlighting the benefit of combining sample-aware aggregation with critic decorrelation for more reliable multi-agent learning.

As reported in Table 1, the proposed SERA framework produces the largest performance improvement in the Multiwalker environment, achieving a gain of 41.1%. Across all evaluated tasks, the method yields an average improvement of 28.7% over the competing approaches. SERA consistently reaches higher cumulative returns, suggesting better utilization of collected samples and improved learning efficiency. Among the baseline algorithms, MASAC provides the strongest competitive performance due to the stability offered by entropy regularization and its twin-critic structure, followed by MATD3 and MAPPO. SERA also has better sample efficiency than the baseline methods. An analysis is given in the Appendix D.1.

## 5.2 Generalization to single agent continuous control

To further examine the generalization capability of SERA, experiments are also conducted on single-agent continuous control tasks, namely Ant, HalfCheetah, and Humanoid. As the multi-agent study already includes widely used CTDE-based methods such as MATD3, MASAC, and MAPPO, their corresponding

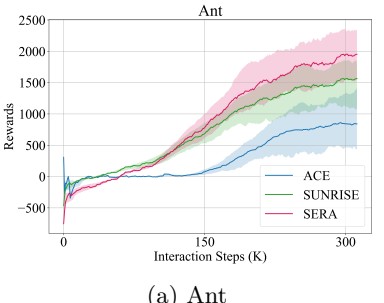 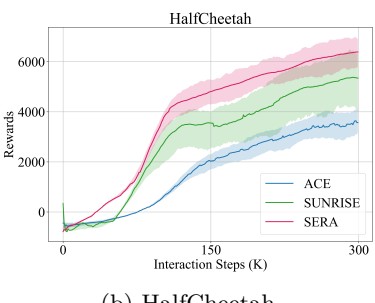 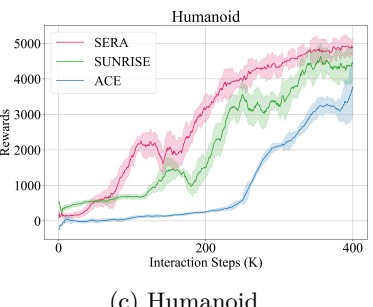

(a) Ant  (b) HalfCheetah  (c) Humanoid

Figure 3: Training reward trajectories comparing the proposed SERA framework with recent ensemble-based baselines across three single-agent MuJoCo environments.

single-agent variants are not considered again in order to avoid repetitive comparisons. Instead, the evaluation focuses on recent ensemble-oriented reinforcement learning approaches, including SUNRISE (Lee et al., 2021) and ACE (Zhang & Yao, 2019). This allows the comparison to remain centered on ensemble-based learning strategies.

From Fig. 3, it is evident that the competing methods obtain relatively similar performance in the *HalfCheetah* and *Humanoid* tasks, where SERA achieves improvements of 17.12% and 15.23%, respectively. A larger performance gain is observed in the *Ant* environment, where the proposed method improves the results by nearly 31.25%.

## 5.3 Discussion

To verify that the observed improvements are not due to random variation, we performed Welch's t-tests using the final evaluation returns across different random seeds. SERA showed statistically significant improvements over MASAC ($p = 0.021$) and IET ($p = 0.029$), indicating consistent gains across runs. The comparison with Ensemble Mean resulted in a larger p-value ($p = 0.072$), suggesting that part of the improvement arises from the variance reduction effect of the ensemble itself. It should also be noted that Ensemble Mean uses the same adaptive learning rate mechanism proposed in SERA.

We also analyze the computational overhead associated with ensemble critics by comparing wall-clock training time against MASAC. Using MASAC as the reference point ($1.0\times$), IET requires about $1.48\times$ more training time, SERA requires approximately $1.6\times$ the training time of MASAC. Even with this additional cost, the overhead remains moderate relative to the gains in training stability and learning efficiency. Overall, the results indicate that the extra computation introduced by SERA is justified by its improved and more reliable training behavior.

**Initial-State Bias:** The initial-state bias can be defined as the difference between the Q-value prediction at the starting state and the empirical discounted return obtained from evaluation rollouts. For an ensemble with $K$ critics, the aggregated value estimate is computed as $V(s_0) = \frac{1}{K} \sum_{k=1}^{K} Q_{\theta_k}(s_0, a_0)$, while the corresponding bias is given by $b = V(s_0) - \sum_t \gamma^t r_t$. Here smaller bias reflects more reliable temporal-difference estimation. Fig. 4a presents the evolution of the initial-state bias during training. SERA consistently maintains the smallest bias, suggesting more stable and accurate value estimation. MADDPG, on the other hand, shows clear overestimation behavior, whereas twin-critic methods such as MATD3 and MASAC display a tendency toward underestimation (Ren et al., 2021; Lyu et al., 2022). These observations suggest that controlling estimation variance helps SERA avoid both overly optimistic and overly pessimistic value estimates during MARL training.

**Ablation Study** Fig. 4b reports a focused ablation study of SERA's two key design choices on the Halfcheetah environment. Disabling the variance-adaptive learning rate (SERA-NoLR) or including the median in the soft aggregation (SERA-Median) consistently degrades performance, indicating that both uncertainty-aware updates and robust anchoring play a critical role in SERA's stability and effectiveness. SERA-Median

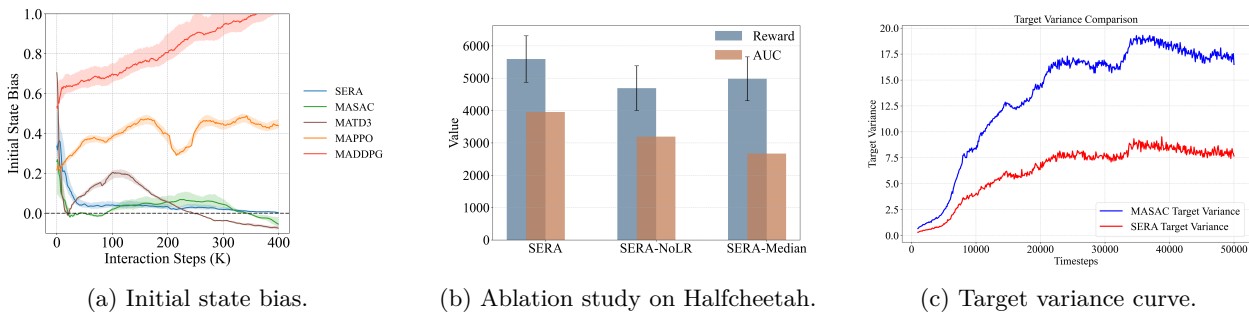

(a) Initial state bias.          (b) Ablation study on Halfcheetah.          (c) Target variance curve.

Figure 4: Experimental analysis of SERA showing initial state bias, ablation results and target variance comparison on Halfcheetah environment.

Table 2: Performance comparison of SERA with ensemble-based baselines on continuous single-agent control tasks.

| Environment | SERA | SUNRISE | ACE |
|---|---|---|---|
| Ant-v5 | **1983.7** $\pm 602$ | $1530 \pm 605$ | $890 \pm 654$ |
| HalfCheetah-v5 | **6056** $\pm 439$ | $5613 \pm 570$ | $3923 \pm 480$ |
| Humanoid-v5 | **4982.6** $\pm 510$ | $4515.3 \pm 600$ | $3908.6 \pm 582$ |

becomes highle conservative as it downweights the other critic values as discussed in section 4.1. In the results section, we compared SERA with Ensemble Mean, which also plays a part of ablation study. Here, in Fig. 4b, the AUC plots are divided by the total number of training episodes to have a better plot.

**Variance Reduction** Figure 4c presents the evolution of TD target variance during training. Compared to MASAC, the proposed SERA approach maintains noticeably lower variance throughout the learning process. The difference becomes larger in the later stages of training, where critic estimates generally become more unstable due to accumulated approximation errors and growing disagreement across critics. The smoother variance profile observed with SERA indicates that the proposed aggregation strategy produces more stable target estimates and reduces the influence of unreliable critic predictions.

**Limitations.** Despite improving the stability of value estimation, SERA requires maintaining multiple critic networks, which increases both training time and computational cost compared to conventional twin-critic approaches. The proposed aggregation mechanism also relies on the ensemble retaining sufficiently different critic estimates during training. If the critics become overly similar, the overall benefit obtained from ensemble-based variance reduction can decrease. In addition, using larger ensembles may further improve robustness, but this comes at the expense of higher memory usage and additional optimization overhead.

# 6 Conclusion

In this work, we investigated the role of critic disagreement and estimation variance in RL, and introduced SERA, a reliability-aware ensemble critic framework for stable target estimation. Instead of relying on hard minimum selection or uniform averaging, SERA constructs targets through a soft reliability-based aggregation centered around the median critic estimate. This allows the framework to suppress unstable or inconsistent critics while avoiding the excessive pessimism often introduced by minimum-based target operators. The proposed method is developed within the centralized training and decentralized execution paradigm and can be integrated into standard off-policy MARL pipelines with minimal modification. Experiments on multiple cooperative continuous-control benchmarks demonstrate that SERA consistently outperforms the strong baseline methods. The results show that reliability-aware target aggregation can serve as an effective alternative to conventional ensemble reduction strategies.

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

# A    Appendix

## A.1    Proofs

### Proof of Remark 1

*Proof.* For a fixed sample $b \in \mathcal{B}$, the denominator of the SERA weight is common to all non-median critics:

$$Z^{(b)} = \sum_{k \neq m_b} \exp\left(-d_k^{(b)}/\tau\right).$$

Since $\tau > 0$, the function

$$f(d) = \exp(-d/\tau)$$

is strictly decreasing in $d$. Therefore, if

$$d_i^{(b)} < d_j^{(b)},$$

then

$$\exp\left(-d_i^{(b)}/\tau\right) > \exp\left(-d_j^{(b)}/\tau\right).$$

Dividing both sides by the same positive denominator $Z^{(b)}$ gives $w_i^{(b)} > w_j^{(b)}$.

This proves the sample-wise ordering.

Now suppose that $d_i^{(b)} \leq d_j^{(b)} \qquad \forall b \in \mathcal{B}$, with strict inequality for at least one sample. By the sample-wise result,

$$w_i^{(b)} \geq w_j^{(b)} \qquad \forall b \in \mathcal{B},$$

and

$$w_i^{(b)} > w_j^{(b)}$$

for at least one sample. Summing over all samples gives

$$\sum_{b \in \mathcal{B}} w_i^{(b)} > \sum_{b \in \mathcal{B}} w_j^{(b)}.$$

Dividing by $|\mathcal{B}|$ yields

$$\frac{1}{|\mathcal{B}|} \sum_{b \in \mathcal{B}} w_i^{(b)} > \frac{1}{|\mathcal{B}|} \sum_{b \in \mathcal{B}} w_j^{(b)}.$$

Hence, a critic that stays closer to the median anchor across the mini-batch receives a larger average SERA reliability weight. □

### Proof of Proposition 4.1

*Proof.* Define

$$\lambda_{\mathrm{med}} = \alpha, \qquad \lambda_i = (1-\alpha)w_i^{(b)}, \quad i \neq m_b.$$

Since $0 < \alpha < 1$ and $w_i^{(b)} \geq 0$, it follows that

$$\lambda_{\mathrm{med}} \geq 0, \qquad \lambda_i \geq 0.$$

In addition,

$$\lambda_{\mathrm{med}} + \sum_{i \neq m_b} \lambda_i = \alpha + (1-\alpha) \sum_{i \neq m_b} w_i^{(b)} = \alpha + (1-\alpha) = 1.$$

Hence, $Q_{\mathrm{SERA}}^{(b)}$ can be written as a convex combination of the target critic estimates.

Because the median $Q_{\text{med}}^{(b)}$ is one of the target critic values, the SERA estimate is a convex combination of the set $\{Q_{\tilde{\theta}_i}^{(b)}\}_{i=1}^{N}$. Therefore, it lies within their convex hull.

Since the critic outputs are scalar, this convex hull reduces to the interval

$$\left[\min_i Q_{\tilde{\theta}_i}^{(b)}, \max_i Q_{\tilde{\theta}_i}^{(b)}\right].$$

Thus,

$$\min_i Q_{\tilde{\theta}_i}^{(b)} \le Q_{\text{SERA}}^{(b)} \le \max_i Q_{\tilde{\theta}_i}^{(b)}.$$

This shows that the aggregation is not forced to select the smallest critic value, as in min-based methods, while at the same time remaining within the range of the ensemble predictions. $\qquad\square$

This result shows that SERA does not only reduce variance at the critic-output level, but also induces a lower-variance bootstrapped TD target. Since critic learning is driven by regression toward this target, the resulting update signal is less noisy than that obtained from a single target critic.

**Proof of Proposition 4.2**

*Proof.* For a fixed sample $b$, let the target critic estimates be $Q_{\tilde{\theta}_i}^{(b)} = y^\star + \varepsilon_i$, where $\mathbb{E}[\varepsilon_i] = 0$ and $\text{Var}(\varepsilon_i) = \sigma^2$. Since the median anchor corresponds to one of the target critic estimates, there exists an index $m_b$ such that $Q_{\text{med}}^{(b)} = Q_{\tilde{\theta}_{m_b}}^{(b)}$. The SERA target can therefore be written as

$$Q_{\text{SERA}}^{(b)} = \alpha Q_{\tilde{\theta}_{m_b}}^{(b)} + (1-\alpha)\sum_{i \ne m_b} w_i^{(b)} Q_{\tilde{\theta}_i}^{(b)}.$$

Consider the weights $(\tilde{w})$ as given in Proposition 4.2,

$$\tilde{w}_i^{(b)} = \begin{cases} \alpha, & i = m_b, \\ (1-\alpha)w_i^{(b)}, & i \ne m_b. \end{cases}$$

Then,

$$Q_{\text{SERA}}^{(b)} = \sum_{i=1}^{N} \tilde{w}_i^{(b)} Q_{\tilde{\theta}_i}^{(b)}. \tag{14}$$

Since $w_i$ are normalized, thus from (4), we can write

$$\sum_{i \ne m_b} w_i^{(b)} = 1.$$

Thus, the effective weights satisfy

$$\sum_{i=1}^{N} \tilde{w}_i^{(b)} = \alpha + (1-\alpha) = 1.$$

Hence, $Q_{\text{SERA}}^{(b)}$ is a convex combination of the critic estimates. Substituting $Q_{\tilde{\theta}_i}^{(b)} = y^\star + \varepsilon_i$ in (14), we obtain

$$Q_{\text{SERA}}^{(b)} = y^\star + \sum_{i=1}^{N} \tilde{w}_i^{(b)} \varepsilon_i. \tag{15}$$

The goal is to bound the variance of the SERA estimate relative to the true target $y^\star$. We achieve this by bounding its mean-squared error (MSE). For any estimator $X$ of $y^\star$,

$$\text{MSE}(X) = \mathbb{E}\left[(X - y^\star)^2\right].$$

Using the bias–variance decomposition,

$$\mathrm{MSE}(X) = \mathrm{Var}(X) + \left(\mathbb{E}[X] - y^\star\right)^2,$$

which implies

$$\mathrm{Var}(X) \le \mathrm{MSE}(X).$$

Therefore, it suffices to establish an upper bound on $\mathrm{MSE}(Q_{\mathrm{SERA}}^{(b)})$.

Using (15) we can write

$$\mathrm{MSE}(Q_{\mathrm{SERA}}^{(b)}) = \mathbb{E}\left[\left(\sum_{i=1}^{N} \tilde{w}_i^{(b)} \varepsilon_i\right)^2\right]. \tag{16}$$

For every realization, the effective weights satisfy $\tilde{w}_i^{(b)} \ge 0$ and $\sum_{i=1}^{N} \tilde{w}_i^{(b)} = 1$. Since the function $f(x) = x^2$ is convex on $\mathbb{R}$, the finite weighted form of Jensen's inequality Boyd & Vandenberghe (2004) states that for any real numbers $x_1, \ldots, x_N$ and nonnegative weights $\lambda_1, \ldots, \lambda_N$ satisfying $\sum_{i=1}^{N} \lambda_i = 1$,

$$f\left(\sum_{i=1}^{N} \lambda_i x_i\right) \le \sum_{i=1}^{N} \lambda_i f(x_i).$$

Taking $\lambda_i = \tilde{w}_i^{(b)}$ and $x_i = \varepsilon_i$ yields

$$\left(\sum_{i=1}^{N} \tilde{w}_i^{(b)} \varepsilon_i\right)^2 \le \sum_{i=1}^{N} \tilde{w}_i^{(b)} \varepsilon_i^2.$$

Taking expectations on both sides and using (16), we get

$$\mathrm{MSE}\left(Q_{\mathrm{SERA}}^{(b)}\right) \le \sum_{i=1}^{N} \mathbb{E}\left[\tilde{w}_i^{(b)} \varepsilon_i^2\right]. \tag{17}$$

Using the covariance identity

$$\mathbb{E}[AB] = \mathbb{E}[A]\mathbb{E}[B] + \mathrm{Cov}(A, B),$$

with $A = \tilde{w}_i^{(b)}$ and $B = \varepsilon_i^2$, we obtain

$$\mathbb{E}\left[\tilde{w}_i^{(b)} \varepsilon_i^2\right] = \mathbb{E}[\tilde{w}_i^{(b)}]\mathbb{E}[\varepsilon_i^2] + \mathrm{Cov}(\tilde{w}_i^{(b)}, \varepsilon_i^2).$$

Since $\mathbb{E}[\varepsilon_i^2] = \sigma^2$, (17) simplifies to

$$\mathrm{MSE}(Q_{\mathrm{SERA}}^{(b)}) \le \sum_{i=1}^{N} \left(\sigma^2 \mathbb{E}[\tilde{w}_i^{(b)}] + \mathrm{Cov}(\tilde{w}_i^{(b)}, \varepsilon_i^2)\right).$$

Rearranging,

$$\mathrm{MSE}(Q_{\mathrm{SERA}}^{(b)}) \le \sigma^2 \sum_{i=1}^{N} \mathbb{E}[\tilde{w}_i^{(b)}] + \sum_{i=1}^{N} \mathrm{Cov}(\tilde{w}_i^{(b)}, \varepsilon_i^2).$$

Because $\sum_{i=1}^{N} \tilde{w}_i^{(b)} = 1$ and for every realization, $\sum_{i=1}^{N} \mathbb{E}[\tilde{w}_i^{(b)}] = 1.$, we get

$$\mathrm{MSE}(Q_{\mathrm{SERA}}^{(b)}) \le \sigma^2 + \sum_{i=1}^{N} \mathrm{Cov}(\tilde{w}_i^{(b)}, \varepsilon_i^2).$$

By the sufficient condition of Proposition 4.2,

$$\sum_{i=1}^{N} \text{Cov}(\tilde{w}_i^{(b)}, \varepsilon_i^2) < 0.$$

Hence,

$$\text{MSE}(Q_{\text{SERA}}^{(b)}) < \sigma^2.$$

Since

$$\text{Var}(Q_{\text{SERA}}^{(b)}) \leq \text{MSE}(Q_{\text{SERA}}^{(b)}),$$

we obtain

$$\text{Var}(Q_{\text{SERA}}^{(b)}) < \sigma^2.$$

Finally, for each individual critic,

$$\text{Var}(Q_{\bar{\theta}_i}^{(b)}) = \text{Var}(y^\star + \varepsilon_i) = \text{Var}(\varepsilon_i) = \sigma^2,$$

since $y^\star$ is deterministic. Therefore,

$$\text{Var}(Q_{\text{SERA}}^{(b)}) < \sigma^2 = \text{Var}(Q_{\bar{\theta}_i}^{(b)}), \qquad \forall i \in \{1, \ldots, N\}.$$

$\square$

## B  Empirical Verification of Assumption $\sum_{i=1}^{N} \text{Cov}(\tilde{w}_i^{(b)}, \varepsilon_i^2) < 0$.

Proposition 4.2 establishes that the variance of the proposed SERA aggregation is bounded by the covariance between the reliability weights and the squared critic errors. Since the critic errors depend on the unknown Bellman target, the sign of the covariance term cannot be verified analytically during learning. Nevertheless, the design of SERA assigns smaller weights to critics that deviate more from the ensemble median, which serves as an observable reliability signal. To examine whether this design indeed induces the required negative covariance, we conduct a controlled synthetic study in which the true target is known exactly, allowing the sufficient condition of Proposition 4.2 to be evaluated directly.

Throughout this appendix, the ground-truth target is fixed as $y^\star = 0$, and the synthetic critic estimates are generated as

$$\bar{Q}_i = y^\star + \epsilon_i, \qquad i = 1, \ldots, K, \tag{18}$$

where $\epsilon_i$ denotes the estimation error of critic $i$. Unless otherwise stated, the critic errors are drawn from the Gaussian distribution $\epsilon_i \sim \mathcal{N}(0, \sigma^2)$, with $\sigma = 1$.

The SERA aggregation follows exactly the formulation used in the proposed Algorithm method of the main manuscript. The median critic receives weight $\alpha = 0.25$, while the remaining critics are assigned reliability weights

$$w_i = \frac{\exp(-d_i/\tau)}{\sum_{j \neq m} \exp(-d_j/\tau)}, \qquad d_i = |\bar{Q}_i - \bar{Q}_{\text{med}}|, \tag{19}$$

resulting in the effective aggregation weights

$$\tilde{w}_m = \alpha, \qquad \tilde{w}_i = (1 - \alpha)w_i, \qquad i \neq m. \tag{20}$$

All covariance estimates are computed from $10^6$ randomly generated synthetic critic ensembles. For each realization, $K$ target critic estimates are generated according to $\bar{Q}_i = y^\star + \epsilon_i$, where the ground-truth target is fixed at $y^\star = 0$ and the critic errors $\epsilon_i$ are drawn from the specified noise distribution. The proposed SERA aggregation is then applied using the same median-centered reliability weighting described in Section 4, and the covariance term $\sum_i \text{Cov}(\tilde{w}_i, \epsilon_i^2)$ is estimated over the complete set of realizations.

### B.1 Verification of the reliability-alignment condition

We first analyze the covariance term appearing in Proposition 4.2,

$$\sum_{i=1}^{K} \mathrm{Cov}(\tilde{w}_i, \epsilon_i^2),$$

under the operating regime considered in SERA.

### B.1.1 Effect of the softmax temperature $\tau$

The ensemble size is fixed to $K = 5$ and the temperature parameter $\tau$ is varied over a wide range.

Table 3: Reliability-alignment term for different values of $\tau$ ($K = 5$, $\alpha = 0.25$, $\sigma = 1$).

| $\tau$ | $\sum_i \mathrm{Cov}(\tilde{w}_i, \epsilon_i^2)$ |
|---|---|
| 0.01 | -0.561813 |
| 0.02 | -0.561458 |
| 0.05 | -0.559044 |
| 0.10 | -0.550897 |
| 0.15 | -0.538719 |
| 0.20 | -0.523764 |
| 0.30 | -0.489711 |
| 0.50 | -0.422003 |
| 0.70 | -0.365972 |
| 1.00 | -0.303905 |
| 2.00 | -0.198383 |
| 5.00 | -0.112332 |
| 10.00 | -0.079439 |

The covariance remains strictly negative throughout the entire temperature range. Smaller values of $\tau$ produce more selective reliability weighting, resulting in a stronger negative covariance, whereas larger temperatures gradually approach uniform weighting.

### B.1.2 Seed robustness

To verify that the observed covariance estimates are consistent across randomly generated realizations of the synthetic critic ensemble, the experiment is repeated using five different random seeds.

Table 4: Seed robustness of the covariance estimate ($K = 5$, $\tau = 0.1$).

| Seed | $\sum_i \mathrm{Cov}(\tilde{w}_i, \epsilon_i^2)$ |
|---|---|
| 0 | -0.550897 |
| 1 | -0.551615 |
| 2 | -0.552810 |
| 3 | -0.550791 |
| 4 | -0.550707 |
| Mean | -0.551364 |
| Std. | $8.85 \times 10^{-4}$ |

The extremely small standard deviation demonstrates that the estimated covariance is highly reproducible.

Table 5: Effect of ensemble size on the covariance term.

| $K$ | $\sum_i \text{Cov}(\tilde{w}_i, \epsilon_i^2)$ |
|---|---|
| 3 | -0.136967 |
| 5 | -0.550897 |
| 7 | -0.695356 |
| 9 | -0.767682 |
| 11 | -0.811857 |

### B.1.3 Effect of ensemble size

The ensemble size is varied while keeping all other parameters fixed.

The magnitude of the negative covariance increases with the ensemble size, indicating that the proposed reliability weighting becomes more effective as additional critic diversity is introduced.

### B.1.4 Distribution robustness

Finally, we evaluate the covariance under different error distributions.

Table 6: Covariance under different critic error distributions.

| Distribution | $\sum_i \text{Cov}(\tilde{w}_i, \epsilon_i^2)$ |
|---|---|
| Gaussian | -0.550897 |
| Laplace | -0.726471 |
| Uniform | -0.339042 |
| Student-$t$ | -0.709832 |

The covariance remains negative for all tested distributions, indicating that the sufficient condition of Proposition 4.2 is not restricted to Gaussian critic errors.

The experiments shown above evaluate the sufficient condition introduced in Proposition 4.2. Unless otherwise specified, all synthetic critics are generated with Gaussian errors of standard deviation $\sigma = 1$.

## B.2 Variance reduction

Having established that the covariance condition holds, we now turn to the more practical setting in which critics have different levels of uncertainty. This setting is more representative of deep ensemble reinforcement learning and enables us to evaluate the empirical variance reduction achieved by SERA. We consider both heterogeneous-noise regime relevant to deep reinforcement learning and the homogeneous-noise regime used as a theoretical control.

### B.2.1 Heterogeneous critic ensemble (primary regime)

To better reflect practical deep reinforcement learning, where critics may differ because of their architectures, random initializations, and training histories (Table 4 of the main text), we consider a heterogeneous uncertainty setting with $(\sigma_1, \ldots, \sigma_5) = (0.5, 1.0, 1.0, 1.5, 3.0)$. In each realization, the critic errors are sampled from their respective Gaussian distributions, after which the critic estimates are aggregated using both uniform averaging and SERA. This process is repeated for $10^6$ random realizations, and the variance of each aggregation method is computed from the resulting estimates. In this setting, the critics no longer have the same level of uncertainty. While uniform averaging treats every critic equally regardless of its reliability, SERA adjusts each critic's contribution based on its agreement with the ensemble median. The results in Table 7 show that SERA achieves lower empirical variance than an individual critic, consistent with Propo-

sition 4.2. They also show that, when critic uncertainties differ, SERA can outperform the uniform ensemble mean, highlighting the benefit of reliability-aware aggregation in heterogeneous ensembles.

### B.2.2 Homogeneous critic ensemble (theoretical control)

As a control condition, we also evaluate the case of i.i.d. critic noise, $\epsilon_i \sim \mathcal{N}(0, 1)$ across critics. Under this assumption, classical estimation theory dictates that the arithmetic mean is the minimum-variance unbiased linear combination, since all critics carry identical, uncorrelated noise; no reliability-aware weighting scheme can improve on uniform averaging here, and SERA does not attempt to. This is consistent with the discussion already given in Section 4.1 of the main text (**"Comparison with mean aggregation"**), where we note that uniform averaging is variance-optimal precisely when critic noise is i.i.d., and that SERA's advantage is expected specifically in the heterogeneous, unevenly-reliable regime that characterizes deep ensemble critics in practice. Table 8 confirms this: SERA still substantially reduces variance relative to a single critic (60.67%), but, as mentioned in section 4.1, does not improve on the uniform mean in this idealized i.i.d. setting.

.

Table 7: Variance comparison under heterogeneous critic uncertainty (primary regime).

| Estimator | Variance |
|---|---|
| Single critic | 2.699684 |
| Uniform mean | 0.539553 |
| SERA | 0.443476 |
| Reduction vs. single | 83.57% |
| Reduction vs. mean | 17.81% |

Table 8: Variance comparison under homogeneous (i.i.d.) critic uncertainty (theoretical control).

| Estimator | Variance |
|---|---|
| Single critic | 0.999780 |
| Uniform mean | 0.200054 |
| SERA | 0.393208 |
| Reduction vs. single | 60.67% |
| Reduction vs. mean | −96.55% |

### B.2.3 Sensitivity to critic heterogeneity

The previous experiment examines a single heterogeneous variance configuration. To determine whether the same behavior persists under different levels of critic heterogeneity, we extend the synthetic study to five representative variance settings, ranging from identical critic uncertainties to ensembles with increasingly diverse uncertainty levels.

**Key observations**: Table 9 highlights a clear trend as the level of critic heterogeneity increases. In the homogeneous setting (H1), where all critics have identical uncertainty, uniform averaging achieves the lowest variance, consistent with the classical result that equal weighting is optimal when all estimators have the same variance.

As the differences in critic uncertainty become more pronounced (H2–H4), the advantage of reliability-aware aggregation becomes evident. SERA consistently achieves lower variance than the ensemble mean, with the reduction increasing from 17.71% in H2 to 59.72% in H4. This trend indicates that the benefit of adaptive weighting grows as the reliability gap between critics widens. By reducing the influence of high-variance critics while preserving contributions from more reliable ones, SERA produces more stable target

Table 9: Extended Synthetic heterogeneous critic variance experiment on multiple sigma sets. Each result is averaged over 10 independent random seeds, with $10^6$ random realizations per seed. Lower variance is better.

| Case | Critic std. deviations $(\sigma_1, \ldots, \sigma_5)$ | $\mathrm{Var}(Q_{\mathrm{Mean}})$ | $\mathrm{Var}(Q_{\mathrm{SERA}})$ | Reduction vs Mean(%) |
|------|-------------------------------------------------------|-----------------------------------|-----------------------------------|----------------------|
| H1 | $(0.5, 0.5, 0.5, 0.5, 0.5)$ | $0.050 \pm 0.000$ | $0.094 \pm 0.000$ | $-87.99 \pm 0.28$ |
| H2 | $(0.5, 1.0, 1.0, 1.5, 3.0)$ | $0.540 \pm 0.001$ | $\mathbf{0.444 \pm 0.001}$ | $\mathbf{17.71 \pm 0.13}$ |
| H3 | $(0.5, 1.0, 2.0, 3.0, 5.0)$ | $1.570 \pm 0.003$ | $\mathbf{0.913 \pm 0.002}$ | $\mathbf{41.85 \pm 0.12}$ |
| H4 | $(0.5, 1.0, 3.0, 5.0, 8.0)$ | $3.971 \pm 0.006$ | $\mathbf{1.600 \pm 0.004}$ | $\mathbf{59.72 \pm 0.11}$ |

estimates than uniform averaging in heterogeneous settings. These results support the motivation for SERA in practical deep reinforcement learning, where ensemble critics naturally differ (Liang et al. (2022)) because of their architectures, random initializations, and optimization histories. In our framework, this diversity is further encouraged through heterogeneous critic architectures, distinct parameter initializations, and the proposed variance-adaptive learning rate, which together promote critic diversity and make reliability-aware aggregation particularly effective.

The synthetic results are consistent with Proposition 4.2. In particular, the reduction in estimator variance observed under heterogeneous critic uncertainties is consistent with the conditions described in Proposition 4.2. When considered alongside the covariance analysis presented earlier, these experiments provide additional evidence that SERA can produce more stable target estimates in the heterogeneous critic settings commonly encountered in deep reinforcement learning.

**Reduced variance of the SERA TD target** : For a fixed transition, define the TD target based on a single critic $j$ as

$$y_j = r + \gamma Q_{\bar{\theta}_j}(s', \mathbf{a}'),$$

and the SERA TD target as

$$y_{\mathrm{SERA}} = r + \gamma Q_{\mathrm{SERA}}(s', \mathbf{a}').$$

If the reward term is fixed for the sampled transition, then $\mathrm{Var}(y_{\mathrm{SERA}}) = \gamma^2 \mathrm{Var}(Q_{\mathrm{SERA}})$ and $\mathrm{Var}(y_j) = \gamma^2 \mathrm{Var}(Q_{\bar{\theta}_j})$. Therefore, by Proposition 4.2, $\mathrm{Var}(y_{\mathrm{SERA}}) < \mathrm{Var}(y_j)$.

**Proof of Lemma 4.3**

*Proof.* Let the estimation error at step $t$ be

$$e_t = Q_t - Q^\star(s, a), \qquad \varepsilon_t = y_t - Q^\star(s, a).$$

Both quantities are assumed to be zero-mean, so that $\mathbb{E}[e_t] = 0$ and $\mathbb{E}[\varepsilon_t] = 0$.

Consider the critic update written as a convex combination of the current estimate and the target:

$$Q_{t+1} = (1 - \beta_t)Q_t + \beta_t y_t.$$

Subtracting $Q^\star(s, a)$ from both sides gives

$$
\begin{aligned}
e_{t+1} &= Q_{t+1} - Q^\star(s, a) \\
&= (1 - \beta_t)Q_t + \beta_t y_t - Q^\star(s, a) \\
&= (1 - \beta_t)\big(Q_t - Q^\star(s, a)\big) + \beta_t\big(y_t - Q^\star(s, a)\big) \\
&= (1 - \beta_t)e_t + \beta_t \varepsilon_t.
\end{aligned}
$$

Squaring both sides yields

$$e_{t+1}^2 = (1 - \beta_t)^2 e_t^2 + \beta_t^2 \varepsilon_t^2 + 2(1 - \beta_t)\beta_t e_t \varepsilon_t.$$

Taking expectations, we obtain

$$\mathbb{E}[e_{t+1}^2] = (1 - \beta_t)^2 \mathbb{E}[e_t^2] + \beta_t^2 \mathbb{E}[\varepsilon_t^2] + 2(1 - \beta_t)\beta_t \mathbb{E}[e_t \varepsilon_t].$$

Since $e_t$ and $\varepsilon_t$ are uncorrelated, the cross term vanishes:

$$\mathbb{E}[e_t \varepsilon_t] = 0.$$

Therefore,

$$\mathbb{E}[e_{t+1}^2] = (1 - \beta_t)^2 \mathbb{E}[e_t^2] + \beta_t^2 \mathbb{E}[\varepsilon_t^2].$$

Using the definitions $P_t = \mathbb{E}[e_t^2]$ and $R_t = \mathbb{E}[\varepsilon_t^2]$, we obtain

$$\mathbb{E}\big[(Q_{t+1} - Q^\star(s, a))^2\big] = (1 - \beta_t)^2 P_t + \beta_t^2 R_t.$$

To obtain the variance-minimizing step size, differentiate with respect to $\beta_t$:

$$\frac{\partial}{\partial \beta_t}\Big[(1 - \beta_t)^2 P_t + \beta_t^2 R_t\Big] = -2(1 - \beta_t)P_t + 2\beta_t R_t.$$

Setting the derivative to zero gives

$$-2(1 - \beta_t)P_t + 2\beta_t R_t = 0,$$

which simplifies to

$$\beta_t^\star = \frac{P_t}{P_t + R_t}.$$

$\square$

**Remark 4** (Common bias)**.** *The zero-mean assumption simplifies the derivation but is not essential. Consider the case where both the critic estimation error and the TD-target error share the same constant bias $b$, i.e.,*

$$e_t = b + \tilde{e}_t, \qquad \varepsilon_t = b + \tilde{\varepsilon}_t,$$

*where*

$$\mathbb{E}[\tilde{e}_t] = \mathbb{E}[\tilde{\varepsilon}_t] = 0, \qquad \mathbb{E}[\tilde{e}_t \tilde{\varepsilon}_t] = 0.$$

*Substituting these expressions into the update equation gives*

$$e_{t+1} = b + (1 - \beta_t)\tilde{e}_t + \beta_t \tilde{\varepsilon}_t.$$

*Taking expectations of the squared error yields*

$$\mathbb{E}[e_{t+1}^2] = b^2 + (1 - \beta_t)^2 P_t + \beta_t^2 R_t,$$

*where*

$$P_t = \mathbb{E}[\tilde{e}_t^2], \qquad R_t = \mathbb{E}[\tilde{\varepsilon}_t^2].$$

*Since the constant term $b^2$ does not depend on $\beta_t$, it does not influence the optimization. Consequently, the optimal step size remains*

$$\beta_t^\star = \frac{P_t}{P_t + R_t}.$$

*Therefore, the derivation does not require the individual errors to be unbiased, provided that the critic estimate and the TD target share the same constant bias. In this case, the optimal update depends on the relative bias between the two quantities rather than on their absolute bias.*

The assumptions used in Lemma 4.3 and the above remark are introduced only to simplify the theoretical analysis. They are not expected to hold exactly under nonlinear function approximation in practical deep reinforcement learning. As discussed in Remark 3, the resulting adaptive step size serves as an uncertainty-aware heuristic, whose effectiveness is demonstrated through the empirical results.

### B.3 Sensitivity Analysis of the Adaptive Learning Rate

In this subsection, we analyze how the adaptive learning rate $\beta_t^\star$ responds to variations in two key sources of uncertainty: uncertainty in the critic's current estimate and uncertainty in the temporal-difference (TD) target. We further discuss how these factors jointly influence the resulting learning behavior.

- **High critic uncertainty, low target uncertainty.** When $P_t$ is large and $R_t$ is small, $\beta_t^\star$ moves closer to one, producing a larger update step. This occurs when the critic estimate is uncertain but the TD target remains reliable, making stronger updates desirable.

- **High critic uncertainty, high target uncertainty.** If both $P_t$ and $R_t$ are large, the update is automatically moderated because the denominator also increases. Although the critic is uncertain, the target itself is noisy, so overly aggressive updates are avoided.

- **Low critic uncertainty, low target uncertainty.** When both uncertainty measures are small, the learning rate stays moderate. In this setting, the critic estimate is already stable and the TD target is reliable, so learning mainly performs gradual refinement.

- **Low critic uncertainty, high target uncertainty.** When $P_t$ is small but $R_t$ is large, the learning rate decreases. Here, the critic estimate is comparatively reliable, whereas the TD target is noisy, and smaller updates help maintain stability.

Overall, the adaptive learning rate does not depend on critic uncertainty alone; rather, it reflects a balance between the trustworthiness of the current value estimate and the reliability of incoming TD targets. This balance enables stable learning while still allowing rapid adaptation when informative and reliable targets are available.

### B.4 Decorrelation strategy

**Relationship between $R_t$ and $R_{\mathbf{gain}}$**: We now clarify the connection between $R_t$ and $R_{\text{gain}}$. For simplicity, we first present the derivation using a generic target critic $Q_{\bar{\theta}}$, and later extend the formulation to each ensemble member $Q_{\bar{\theta}_i}$.

For a transition $(s, a, r, s')$, $y = r + \gamma Q_{\bar{\theta}}(s', a')$, denotes TD target for training, where $Q_{\bar{\theta}}(s', a')$ is the target estimate (used by DDPG, SAC and TD3).

Bellman optimality equation is,

$$Q^\star(s, a) = \mathbb{E}[r + \gamma Q^\star(s', a') \mid s, a],$$

where $a' = \pi^\star(s') = \arg\max_{\bar{a}} Q^\star(s', \bar{a})$.

Now, subtracting $Q^\star(s, a)$ from $y$, and by addition and subtraction of $\gamma Q^\star(s', a')$ in the RHS, we get:

$$y - Q^\star(s, a) = \Big(r + \gamma Q^\star(s', a') - Q^\star(s, a)\Big) + \gamma\Big(Q_{\bar{\theta}}(s', a') - Q^\star(s', a')\Big).$$

Let us define the first term as Bellman sampling noise and take variance on both sides,

$$\text{Var}\Big(y_t - Q^\star(s, a)\Big) = \text{Var}(\epsilon_B) + \gamma^2 \, \text{Var}\Big(Q_{\bar{\theta}}(s', a')) - Q^\star(s', a')\Big) + 2\gamma \, \text{Cov}\Big(\epsilon_B, Q_{\bar{\theta}}(s', a') - Q^\star(s', a')\Big).$$

In practice, the Bellman sampling noise and the target estimation error are expected to have only limited correlation. Therefore, the covariance term is treated as sufficiently small and is ignored to simplify the analysis. The first term reflects transition noise and is independent of the critic. On ignoring the scaling factor $\gamma^2$, we get

$$R_t := \text{Var}\big(y_t - Q^\star(s, a)\big) \approx \text{Var}\big(Q_{\bar{\theta}}(s', a') - Q^\star(s', a')\big). \tag{2}$$

Since $Q^\star(s', a')$ is not known, we approximate it using the ensemble-based estimate $Q_{\text{SERA}}(s', a')$ introduced in Section 5.1 as a stable target, consistent with the DRL ensemble papers like MeanQ, EMAX and others. Thus,

$$R_t \approx \text{Var}\left(Q_{\bar{\theta}}(s', a') - Q_{\text{SERA}}(s', a')\right). \tag{3}$$

This equation (3) matches the formulation of $R_{\text{gain}}$. Thus $R_{\text{gain}}$ serves as the practical proxy of $R_t$, that is, for each critic $i$, we estimate the measurement variance as

$$R_{\text{gain},i} = \text{Var}_{j \in \mathcal{B}}\left(Q_{\bar{\theta}_i}(s'_j, \mathbf{a}'_j) - Q_{\text{SERA}}(s'_j, \mathbf{a}'_j)\right), \tag{21}$$

where $\mathcal{B}$ denotes the mini-batch. Here, $Q_{\text{SERA},j}$ serves as the aggregated target, and the deviation from it reflects the critic's estimation error.

**Likelihood motivation and empirical support**   Since the true value $Q^\star(s', a')$ is not directly available during training, we motivate the use of $Q_{\text{SERA}}(s', a')$ as a reliability-weighted proxy from a likelihood perspective. For a fixed next state–action pair $(s', a')$, assume that each target critic provides a noisy estimate of the true value:

$$Q_{\bar{\theta}_i}(s', a') = Q^\star(s', a') + \xi_i, \qquad \xi_i \sim \mathcal{N}(0, \sigma_i^2),$$

where $\sigma_i^2$ represents the uncertainty associated with critic $i$.

Let $q$ denote a candidate estimate of $Q^\star(s', a')$. Under the Gaussian noise assumption, the likelihood of observing the ensemble predictions is

$$p\left(\{Q_{\bar{\theta}_i}(s', a')\}_{i=1}^N \mid q\right) = \prod_{i=1}^N \frac{1}{\sqrt{2\pi\sigma_i^2}} \exp\left(-\frac{\left(Q_{\bar{\theta}_i}(s', a') - q\right)^2}{2\sigma_i^2}\right).$$

Maximizing this likelihood is equivalent to minimizing the negative log-likelihood:

$$\mathcal{L}(q) = \sum_{i=1}^N \frac{\left(Q_{\bar{\theta}_i}(s', a') - q\right)^2}{2\sigma_i^2}.$$

Differentiating with respect to $q$ gives

$$\frac{\partial \mathcal{L}(q)}{\partial q} = \sum_{i=1}^N \frac{q - Q_{\bar{\theta}_i}(s', a')}{\sigma_i^2}.$$

Setting the derivative to zero,

$$\sum_{i=1}^N \frac{q - Q_{\bar{\theta}_i}(s', a')}{\sigma_i^2} = 0,$$

which yields

$$q \sum_{i=1}^N \frac{1}{\sigma_i^2} = \sum_{i=1}^N \frac{Q_{\bar{\theta}_i}(s', a')}{\sigma_i^2}.$$

Therefore, the maximum-likelihood estimate is

$$\hat{q}_{\text{MLE}} = \frac{\sum_{i=1}^N \sigma_i^{-2} Q_{\bar{\theta}_i}(s', a')}{\sum_{i=1}^N \sigma_i^{-2}}.$$

Thus, under Gaussian critic noise, the likelihood-optimal estimate of $Q^\star(s', a')$ becomes an inverse-uncertainty weighted combination of the target critic estimates.

In practice, the critic-specific variances $\sigma_i^2$ are unknown and cannot be measured during training. Although inverse-variance weighting is the likelihood-optimal estimator under Gaussian critic noise, it cannot be applied directly because these variances are unavailable. SERA therefore does not attempt to approximate inverse-variance weighting. Instead, it estimates critic reliability from the observable deviation of each critic from the ensemble median.

To motivate this reliability measure, we consider the same local Gaussian error model introduced above,

$$Q_{\bar{\theta}_i}(s', a') = Q^\star(s', a') + \xi_i, \qquad \xi_i \sim \mathcal{N}(0, \sigma_i^2),$$

where $Q^\star(s', a')$ denotes the true target value, $\xi_i$ is the estimation error of critic $i$, and $\sigma_i^2$ represents its uncertainty. This model is used only to provide intuition for the proposed reliability measure and is not intended to describe the behavior of deep neural network critics during training. Since the true target value is unavailable in practice (Remark 1), SERA uses the ensemble median as a robust reference. Assuming that the ensemble median is locally concentrated around the true target,

$$Q_{\text{med}} \approx Q^\star,$$

the deviation from the median can be approximated as

$$d_i = |Q_i - Q_{\text{med}}| \approx |Q_i - Q^\star| = |\xi_i|.$$

Because $\xi_i$ follows a zero-mean Gaussian distribution, the standard identity

$$\mathbb{E}[|\xi_i|] = \sqrt{\frac{2}{\pi}} \, \sigma_i$$

implies that

$$\mathbb{E}[d_i] \approx \sqrt{\frac{2}{\pi}} \, \sigma_i.$$

Hence, under this local Gaussian approximation, critics with larger uncertainty are expected to exhibit larger deviations from the ensemble median, making the median deviation a natural observable proxy for critic reliability.

To further investigate this relationship, we conducted a synthetic experiment using heterogeneous critic ensembles. The goal of this study is not to mimic the behavior of deep neural network critics, but to examine whether the proposed reliability measure exhibits the expected ordering under the same local Gaussian error model used in the preceding motivation. We considered ten heterogeneous critic configurations by assigning different standard deviations to the five critics. For each configuration, $10^6$ synthetic realizations of the critic estimates were generated across 10 random seeds according to the Gaussian error model. For every realization, we computed the ensemble median, the corresponding median deviation,

$$d_i = |Q_i - Q_{\text{med}}|,$$

and the effective SERA weight for each critic. Unlike the analytical argument above, this experiment does not assume that $Q_{\text{med}} \approx Q^\star$. Instead, the ensemble median is computed directly from the simulated critic estimates in every realization.

Figure 5 summarizes the results of the synthetic study. Across all ten heterogeneous critic configurations, critics with higher uncertainty consistently show larger deviations from the ensemble median and, in turn, receive smaller effective SERA weights. This behavior is consistent with the analytical motivation presented earlier and suggests that the median deviation retains the relative ordering of critic uncertainty.

These results should not be interpreted as evidence that the deviation from the ensemble median approximates the inverse-variance weighting derived from the Gaussian likelihood model. Rather, they support its use as an observable indicator of critic reliability when the critic-specific variances are unavailable during training. Although the resulting weights are based on a different observable quantity, they preserve the observed ordering of critic reliability by assigning larger weights to more reliable critics and smaller weights to

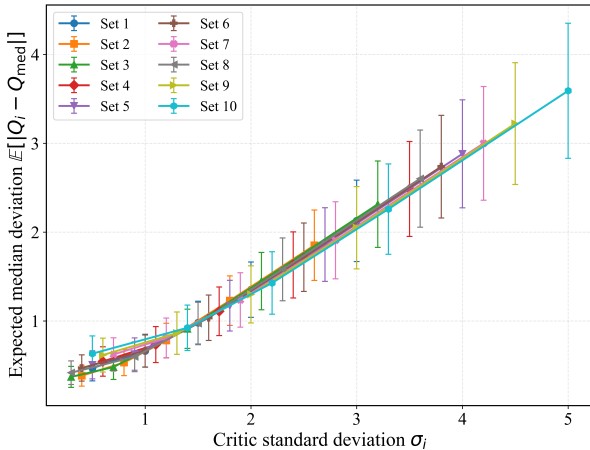 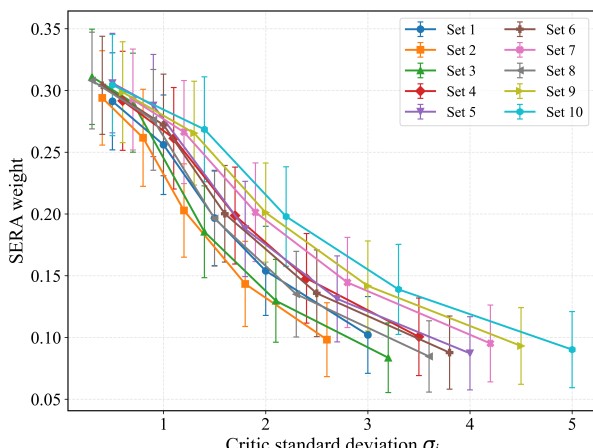

(a) Expected median deviation versus critic standard deviation across ten heterogeneous critic configurations.

(b) Mean effective SERA weight versus critic standard deviation across the same configurations.

Figure 5: Illustration of the relationship between critic uncertainty, median deviation, and the effective SERA weights. Each curve represents one heterogeneous critic configuration. In (a), the expected deviation from the ensemble median increases consistently as the critic uncertainty increases. In (b), the corresponding effective SERA weight decreases with increasing critic uncertainty. Each marker shows the average value obtained over 10 independent runs, with each run computed from $10^6$ independently generated critic estimates. The vertical error bars represent the within-run standard deviation/variability from the $10^6$ randomly generated synthetic critic estimates. Across the ten configurations, the mean Pearson/Spearman correlations between the critic standard deviation and the expected median deviation were $0.994 \pm 0.001$ and $1.000$, respectively. For the effective SERA weight, the corresponding correlations were $-0.981 \pm 0.005$ and $-1.000$.

less reliable ones. This behavior is consistent with the central idea behind inverse-variance weighting, namely that more reliable estimates should contribute more strongly to the final prediction. The MLE derivation is therefore included only to illustrate the ideal estimator when the critic variances are known, whereas the proposed weighting scheme serves as a **practical heuristic** for the setting in which these variances cannot be observed.

**Connection between $P_t$ and TD residual variance.** The ideal estimation variance is defined as

$$P_t = \mathrm{Var}\big(Q_t - Q^\star(s, a)\big),$$

which measures the uncertainty in the critic's current value estimate. In temporal-difference learning, each critic is updated by comparing its current estimate against its corresponding bootstrapped target network. Therefore, the uncertainty of an individual critic is naturally reflected through the variability of its own TD residuals. Since the true value $Q^\star(s, a)$ is unknown, this quantity cannot be computed directly during training.

Consider the temporal-difference residual

$$\delta_t = y_t - Q_t,$$

where $y_t$ denotes the bootstrapped TD target. When the target estimate is approximately unbiased, we have

$$y_t \approx Q^\star(s, a).$$

Substituting this into the TD residual gives

$$\delta_t = y_t - Q_t \approx Q^\star(s, a) - Q_t.$$

Therefore,

$$\mathrm{Var}(\delta_t) \approx \mathrm{Var}\big(Q_t - Q^\star(s, a)\big) = P_t.$$

This shows that the variance of TD residuals can be used as an empirical proxy for the estimation variance. Accordingly, for critic $i$, we estimate this quantity over a mini-batch using

$$v_{Q,i}^{(t)} \leftarrow \alpha_Q v_{Q,i}^{(t-1)} + (1 - \alpha_Q) \mathrm{Var}_{j \in \mathcal{B}} \left( \delta_j^{(i)} \right).$$

### B.5 Effect of Adaptive Learning Rates on Inter-Critic Correlation

Figure 6 shows the correlation patterns among critic heads for both fixed and adaptive learning rates, in the early phase of training. When a fixed learning rate is used, several critic pairs become highly correlated, in some cases with correlation values above 0.9, suggesting that the ensemble members tend to evolve in a strongly synchronized manner.ncy. With the adaptive learning rate, the critic correlation pattern becomes more balanced, reducing excessively strong correlations while maintaining overall agreement among critics. This suggests that the adaptive update mechanism helps prevent critics from becoming overly coupled, allowing them to follow the target values more independently instead of converging to nearly identical estimates. Although the correlations remain positive, they are less tightly clustered near one, indicating lower synchronization while still preserving consistent learning behavior.

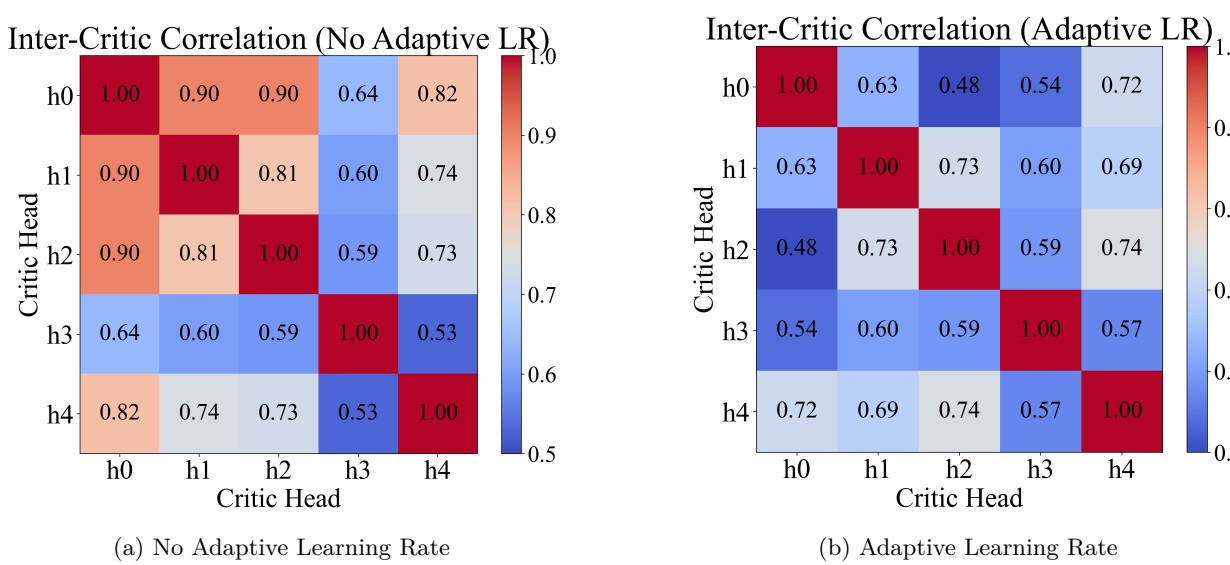

(a) No Adaptive Learning Rate         (b) Adaptive Learning Rate

Figure 6: Inter-critic correlation matrices for a five-head ensemble critic after 50k timesteps. Critics exhibit strong correlation and near-duplicate behavior when the adaptive learning rate is not used.

## C   Algorithm requirements

**Actor Update:** While $Q_{\mathrm{SERA}}$ is employed to construct a stable and variance-controlled target for critic learning, the policy network is updated using the median ensemble estimate $Q_{\mathrm{med}}$. The requirements of critic learning and actor optimization are fundamentally different. Critic updates mainly benefit from reduced target variance, whereas actor updates require a reliable optimization signal that is less affected by temporary estimation errors within the ensemble. Directly optimizing the actor with the adaptive SERA aggregation can encourage the policy to exploit critics that momentarily receive larger weights during training. Using the median estimate alleviates this issue by providing a robust central estimate of the ensemble predictions. At the same time, it avoids the excessive conservatism often introduced by minimum-based operators while remaining resistant to outlier critic estimates.

Using different aggregation strategies for critic targets and actor updates is also common in ensemble reinforcement learning methods. For instance, TD3 Fujimoto et al. (2018) constructs critic targets using the clipped minimum of two critics, whereas the actor is optimized using a single critic estimate. Similarly, SUNRISE Lee et al. (2021) incorporates ensemble uncertainty into critic target estimation through weighted

Bellman backups, while policy optimization is handled separately from the uncertainty-aware aggregation process. Motivated by these observations, SERA uses variance-aware aggregation to stabilize critic learning and employs the median ensemble estimate to provide a robust and stable signal for actor optimization.

## C.1 Hyper-parameters and Network Architectures

In this section, we present the network architectures and training hyperparameters used in the simulation study, which form the basis for all experimental evaluations.

### C.1.1 Network Architectures

Table 10 summarizes the network designs used for each algorithm. Here, "$H \times U$" denotes $H$ hidden layers with $U$ units per layer.

Table 10: Network designs used for each algorithm.

| Algorithm | #Actors | #Critics | Ensemble | Actor MLP | Critic MLP | Activation |
|---|---|---|---|---|---|---|
| MATD3 | $N$ | $2N$ (twin critic) | 1 | $2 \times 256$ | $2 \times 256$ | ReLU |
| MASAC | $N$ | $2N$ (twin critic) | 1 | $2 \times 256$ | $2 \times 256$ | ReLU |
| MAPPO | $N$ (shared/sep) | $N$ (shared/sep) | 1 | $2 \times 128$ | $2 \times 128$ | Tanh / ReLU |
| Ensemble Mean | $N$ | $N$ | 5 | $2 \times 256$ | – | ReLU |
| SERA (ours) | $N$ | $N$ | 5 | $2 \times 256$ | – | ReLU |

**Notes:** $N$ denotes the number of agents. For MAPPO, actor and critic networks may be shared or separate depending on the update strategy. Ensemble sizes are shown as $K$=5 and can be adjusted depending on the experimental setup. To implement IET, we followed the same hyper-parameters used by Shen & How (2023).

### C.1.2 Ensemble Critic Architectures

For Ensemble Mean and SERA, we employ heterogeneous ensemble critics with $K$=5. All critics use ReLU activations between hidden layers and linear output heads. The architectural diversity is summarized in Table 11.

Table 11: Ensemble critic architectures ($K$=5).

| Critic | Hidden Layers | Notes |
|---|---|---|
| $Q^{(1)}$ | $2 \times 256$ | Baseline width |
| $Q^{(2)}$ | 128–256–128 | Deeper, tapered |
| $Q^{(3)}$ | 256–128 | Asymmetric, narrower tail |
| $Q^{(4)}$ | 512–256 | Wider front layer |
| $Q^{(5)}$ | $3 \times 128$ | Deeper, uniformly narrow |

### C.1.3 Training Hyper-parameters

Table 12 reports the key training hyper-parameters used across all algorithms.

Table 12: Key training hyper-parameters.

| Algorithm | LR (A/C) | $\gamma$ | $\tau$ | Batch | Buffer |
|---|---|---|---|---|---|
| MATD3 | $3 \times 10^{-4}/3 \times 10^{-4}$ | 0.99 | 0.005 | 256 | $10^6$ |
| MASAC | $3 \times 10^{-4}/3 \times 10^{-4}$ | 0.99 | 0.005 | 256 | $10^6$ |
| MAPPO | $3 \times 10^{-4}/3 \times 10^{-4}$ | 0.99 | – | 2048 (traj) | – |
| Ensemble Mean | $3 \times 10^{-4}/3 \times 10^{-4}$ | 0.99 | 0.005 | 256 | $10^6$ |
| SERA (ours) | adaptive (variance aware) | 0.99 | 0.005 | 256 | $10^6$ |

We did not compare with **EMAX** as it is not tested in continuous task and also serves different purpose of learning. EMAX was proposed for better exploration.

### C.1.4 Variance-Adaptive Learning Rate Rescaling

To prevent the variance-adaptive gain from collapsing critic updates under high uncertainty, we rescale the effective learning rate. The per-critic gain is defined as $\kappa_t^{(i)}$ given by (11), which would otherwise yield an effective learning rate $\eta_i = \alpha_Q \kappa_t^{(i)} \to 0$ as $\kappa_t^{(i)} \to 0$.

To avoid vanishing updates, we apply a design mapping:

$$\tilde{\kappa}_t^{(i)} = \kappa_{\min} + (1 - \kappa_{\min})\kappa_t^{(i)}, \tag{22}$$

with $\kappa_{\min} = 0.2$, and define the critic step size as $\eta_i = \alpha_{\max}\tilde{\kappa}_t^{(i)}$.

Setting $\alpha_{\max} = 5 \times 10^{-4}$ yields $\eta_i \in [10^{-4}, 5 \times 10^{-4}]$. When uncertainty is low ($\kappa_t^{(i)} \approx 1$), critics take a full update, while under high uncertainty ($\kappa_t^{(i)} \approx 0$), they still perform conservative but non-negligible updates. The actor learning rate is fixed at $3 \times 10^{-4}$ across all experiments.

---

**Algorithm 1** Soft Ensemble Reliability Aggregation (SERA)

---

1: **Initialize:** $M$ actors $\pi_{\phi_m}$, ensemble critics $Q_{\theta^{(i)}}$, target critics $\bar{Q}_{\theta^{(i)}}$, replay buffer $\mathcal{D}$, and per-head estimation variances $P_0^{(i)}$

2: **for** each training step **do**

3:      Sample minibatch $\{(s_j, \mathbf{a}_j, r_j, s'_j, \gamma_j)\}_{j=1}^{S}$ from $\mathcal{D}$

4:      Sample target actions $a'_{j,m} \sim \bar{\pi}_{\phi_m}(\cdot|o'_{m,j})$, form $\mathbf{a}'_j$, and compute $\ell'_j = \sum_m \log \bar{\pi}_{\phi_m}(a'_{j,m}|o'_{m,j})$

5:      **SERA: Reliability-aware target aggregation**

6:      **for** each sample $j = 1, \ldots, S$ **do**

7:          Compute target critic values: $q_j^{(i)} = \bar{Q}_{\theta^{(i)}}(s'_j, \mathbf{a}'_j), \qquad i = 1, \ldots, N.$

8:          Compute median anchor and its index:

$$Q_{\text{med},j} = \text{median}\left(\{q_j^{(i)}\}_{i=1}^{N}\right), \qquad m_j = \text{median\_index}\left(\{q_j^{(i)}\}_{i=1}^{N}\right).$$

9:          Compute median-based deviations: $d_j^{(i)} = \left|q_j^{(i)} - Q_{\text{med},j}\right|.$

10:         Assign soft reliability weights to non-median critics:

$$w_j^{(i)} = \frac{\exp\left(-d_j^{(i)}/\tau_{\text{sera}}\right)}{\sum_{k \neq m_j} \exp\left(-d_j^{(k)}/\tau_{\text{sera}}\right)}, \qquad i \neq m_j.$$

11:         Aggregate the non-median critics: $Q_{\text{sera}\backslash\text{med},j} = \sum_{i \neq m_j} w_j^{(i)} q_j^{(i)}.$

12:         Construct the SERA target estimate: $Q_{\text{SERA},j} = \alpha Q_{\text{med},j} + (1 - \alpha)Q_{\text{sera}\backslash\text{med},j}.$

13:         Construct TD target: $y_j = r_j + \gamma_j \left(Q_{\text{SERA},j} - \alpha_{\text{ent}}\ell'_j\right).$

14:      **end for**

15:      **Variance-adaptive critic updates**

16:      **for** each critic $i = 1, \ldots, N$ **do**

17:         Estimate target uncertainty: $R_{\text{gain},i} = \text{Var}_{j \in B}\left(\bar{Q}_{\theta^{(i)}}(s'_j, \mathbf{a}'_j) - Q_{\text{SERA},j}\right).$

18:         Compute TD error: $\delta_j^{(i)} = y_j - Q_{\theta^{(i)}}(s_j, \mathbf{a}_j).$

19:         Track prediction variance using TD residuals: $v_{Q,i}^{(t)} \leftarrow \alpha_Q v_{Q,i}^{(t-1)} + (1 - \alpha_Q) \text{Var}_{j \in B}\left(\delta_j^{(i)}\right).$

20:         Update estimation variance: $P_t^{(i)} \leftarrow \alpha_P P_{t-1}^{(i)} + (1 - \alpha_P)v_{Q,i}^{(t)}.$

21:         Compute adaptive update scale:

$$\kappa_t^{(i)} = \frac{P_t^{(i)}}{P_t^{(i)} + R_{\text{gain},i} + \eta}.$$

22:         Update critic:

$$\theta^{(i)} \leftarrow \theta^{(i)} + \frac{\alpha_Q}{S} \sum_{j=1}^{S} \kappa_t^{(i)} \delta_j^{(i)} \nabla_{\theta^{(i)}} Q_{\theta^{(i)}}(s_j, \mathbf{a}_j).$$

23:      **end for**

24:      Compute centralized value estimate for actor update: $Q_{\text{cent}}(s, \mathbf{a}) = \text{median}_i Q_{\theta^{(i)}}(s, \mathbf{a}).$

25:      **for** each actor $m = 1, \ldots, M$ **do**

26:         Update actor:

$$\nabla_{\phi_m} J(\phi_m) = \frac{1}{S} \sum_{j=1}^{S} \nabla_{\phi_m} \left[\alpha_{\text{ent}} \log \pi_{\phi_m}(a_{m,j}|o_{m,j}) - Q_{\text{cent}}(s_j, \mathbf{a}_j)\right].$$

27:      **end for**

28:      Soft update target networks

29: **end for**

---

# D    Additional Results and Statistical Analysis

This section presents additional experimental results. We first evaluate the sample efficiency of the compared methods using the normalized area under the learning curve (AUC), followed by an extended statistical significance analysis.

## D.1    Sample Efficiency via Normalized AUC

From the Table 1 it is evident that MATD3 and MAPPO are least comparable so we did not conduct extra experiments for them. Here in this section, we show the normalized AUC for the best three for multi-agent case.

Table 13: Normalized area under the learning curve (AUC) across the multi-agent benchmark environments. Higher values indicate better sample efficiency. Results are reported as mean $\pm$ standard deviation over 10 random seeds.

| Environment | SERA | Ensemble Mean | IET | MASAC |
|---|---|---|---|---|
| HalfCheetah (6 agents) | $\mathbf{3879.2 \pm 383.3}$ | $3462.5 \pm 437.5$ | $3308 \pm 420.8$ | $3139.6 \pm 362.5$ |
| Pusher (3 agents) | $\mathbf{-34.25 \pm 8.25}$ | $-43.33 \pm 7.54$ | $-64.50 \pm 10.42$ | $-48.08 \pm 6.29$ |
| Hopper (3 agents) | $\mathbf{1030.8 \pm 219.6}$ | $817.1 \pm 201.6$ | $670.0 \pm 201.9$ | $622.5 \pm 222.5$ |
| Multi-Walker (3 agents) | $\mathbf{-47.96 \pm 7.46}$ | $-49.12 \pm 7.79$ | $-64.38 \pm 10.83$ | $-54.2 \pm 8.96$ |
| Simple Spread (3 agents) | $\mathbf{-13.10 \pm 1.39}$ | $-14.13 \pm 2.10$ | $-15.80 \pm 2.01$ | $-16.57 \pm 1.53$ |

Table 14: Normalized area under the learning curve (AUC) across the single benchmark environments. Higher values indicate better sample efficiency. Results are reported as mean $\pm$ standard deviation over 10 random seeds.

| Environment | SERA | SUNRISE | ACE |
|---|---|---|---|
| HalfCheetah | $\mathbf{3816.7 \pm 412.3}$ | $3304.2 \pm 333.3$ | $1879 \pm 392.8$ |
| Humanoid | $\mathbf{2912.4 \pm 428.25}$ | $2193.8 \pm 296.92$ | $943.8 \pm 204.42$ |
| Ant | $\mathbf{892.5 \pm 194.6}$ | $797.6 \pm 112.6$ | $253.0 \pm 91.8$ |

Table 13 and Table 14 report the normalized area under the learning curve (AUC), which captures the cumulative return accumulated throughout training and serves as a measure of sample efficiency. SERA consistently achieves the highest normalized AUC across both the multi-agent and single-agent benchmarks, indicating that it learns more effectively during the course of training rather than only at convergence. These results complement the final reward comparisons and suggest that the proposed reliability-aware aggregation improves not only the final policy performance but also the efficiency with which it is learned.

## D.2    Significance Tests

To complement the empirical results, we performed an environment-wise statistical analysis using Welch's t-test on 10 independent random seeds, with Bonferroni correction applied to account for multiple pairwise comparisons. Statistical significance is reported for both the final evaluation return and the normalized area under the learning curve (AUC). The final evaluation return measures the performance attained at the end of training, whereas the normalized AUC summarizes performance over the entire training process and provides an indication of sample efficiency. Considering both metrics offers a more comprehensive evaluation of the proposed method. Table 15 reports the normalized AUC significance results for the five-seed experiments corresponding to the reward-based significance analysis presented in the original manuscript.

We then repeated the analysis using 10 independent random seeds. The results for the multi-agent environments are reported in Tables 16 and 17. In particular, for environments with higher training variability, such

as 3-Agent Hopper and Multi-Walker, the improvement in the final evaluation return achieved by SERA is statistically significant. The normalized AUC analysis further shows that SERA maintains statistically significant gains in sample efficiency across these environments. The corresponding significance results for the single-agent benchmarks are presented in Tables 18 and 19.

Table 15: Environment-wise Welch's $t$-test on the normalized area under the learning curve (AUC) using the same five random seeds as the original manuscript, providing a complementary significance analysis to the reward-based results reported therein. Bonferroni correction was applied independently within each environment for the three pairwise comparisons ($\alpha = 0.0167$).

| Environment | MASAC | IET | Ensemble Mean |
|---|---|---|---|
| 3-Agent Pusher | 0.0066* | 0.0062* | 0.014* |
| 3-Agent Hopper | 0.0090* | 0.0082* | 0.0107* |
| 6-Agent HalfCheetah | 0.0071* | 0.0081* | 0.0123* |
| Simple Spread | 0.0083* | 0.0092* | 0.0161* |
| Multi-Walker | 0.0041* | 0.0042* | 0.0114* |

\* Statistically significant after Bonferroni correction ($\alpha = 0.0167$).

Table 16: Environment-wise Welch's $t$-test (10 random seeds) on the final evaluation return. Bonferroni correction was applied independently within each environment for the three pairwise comparisons ($\alpha = 0.0167$).

| Environment | MASAC | IET | Ensemble Mean |
|---|---|---|---|
| 3-Agent Pusher | 0.0133* | 0.0142* | 0.0431 |
| 3-Agent Hopper | 0.0106* | 0.0119* | 0.0151* |
| 6-Agent HalfCheetah | 0.0023* | 0.0092* | 0.0293 |
| Simple Spread | 0.0092* | 0.0012* | 0.0351 |
| Multi-Walker | 0.0101* | 0.0121* | 0.0135* |

\* Statistically significant after Bonferroni correction ($\alpha = 0.0167$).

Table 17: Environment-wise Welch's $t$-test (10 random seeds) on the normalized area under the learning curve (AUC). Bonferroni correction was applied independently within each environment for the three pairwise comparisons ($\alpha = 0.0167$).

| Environment | MASAC | IET | Ensemble Mean |
|---|---|---|---|
| 3-Agent Pusher | 0.0031* | 0.0052* | 0.0071* |
| 3-Agent Hopper | 0.0010* | 0.0032* | 0.0112* |
| 6-Agent HalfCheetah | 0.0052* | 0.0071* | 0.0093* |
| Simple Spread | 0.0022* | 0.0042* | 0.0153* |
| Multi-Walker | 0.0011* | 0.0012* | 0.0014* |

\* Statistically significant after Bonferroni correction ($\alpha = 0.0167$).

Table 18: Environment-wise Welch's $t$-test (10 random seeds) on the reward curve. Bonferroni correction was applied independently within each environment for the two pairwise comparisons ($\alpha = 0.025$).

| Environment | SUNRISE | ACE |
|---|---|---|
| Ant-v5 | 0.0120* | 0.0100* |
| HalfCheetah-v5 | 0.0190* | 0.0160* |
| Humanoid-v5 | 0.0313 | 0.0013* |

* Significant after Bonferroni correction ($\alpha = 0.025$).

Table 19: Environment-wise Welch's $t$-test (10 random seeds) on the normalized area under the learning curve (AUC) for the single-agent benchmarks. Bonferroni correction was applied independently within each environment for the two pairwise comparisons ($\alpha = 0.025$).

| Environment | SUNRISE | ACE |
|---|---|---|
| Ant-v5 | 0.0022* | 0.0001* |
| HalfCheetah-v5 | 0.0092* | 0.0053* |
| Humanoid-v5 | 0.0031* | 0.0023* |

* Statistically significant after Bonferroni correction ($\alpha = 0.025$).

# E    Behavior of SERA with an Informative Outlier

The reliability weighting mechanism in SERA is motivated by the observation that critics whose predictions remain close to the ensemble consensus tend to produce more accurate value estimates during training. Since the true target value is unknown, it is not possible to determine whether a critic exhibiting a large deviation is genuinely more informative or simply affected by estimation error. Consequently, SERA does not attempt to identify the "correct" critic. Instead, it uses agreement with the ensemble as a practical indicator of reliability when constructing the target estimate.

To further understand this behavior, we perform a mixed-restart experiment in which one critic is initialized from a pretrained checkpoint while the remaining critics are randomly initialized just like when learning starts. Immediately after the restart, the pretrained critic produces value estimates that differ noticeably from those of the other critics. Figure 7(a) shows the behavior of the standard MASAC minimum operator. During the early stages of training, the pretrained critic is selected less frequently because its higher value estimate and the min-operator will choose the minimum one to avoid overestimation. As training continues, however, the remaining critic gradually move toward similar value estimates, and the pretrained critic is selected more often, indicating that the initial disagreement diminishes over time. Thus, the min operators used in MATD3 or MASAC, do not have any knowledge of whether a critic is informative or not.

Figure 7(b) shows the corresponding behavior of SERA under the same setting. During the initial training phase, the pretrained critic receives a relatively small reliability weight because its predictions differ from those of the rest of the ensemble. As the critics become more consistent, the reliability weight assigned to the pretrained critic increases steadily. This behavior highlights an important property of SERA: unlike MASAC critics that temporarily disagree with the ensemble are not discarded. Instead, their influence is reduced only while the disagreement persists and is restored automatically as their estimates become consistent with those of the remaining critics.

In comparison, the ensemble mean treats all critics equally throughout training. With five critics, each contributes a fixed weight of 0.2, regardless of whether the critic is the pretrained one providing potentially useful estimates or a randomly initialized critic that is still learning. The same fixed weighting is retained even if a critic temporarily becomes an unreliable outlier due to estimation errors. As a result, the contribution of each critic remains unchanged during training. SERA, on the other hand, updates the weights at every target computation, allowing the influence of individual critics to increase or decrease as their agreement with the rest of the ensemble changes.

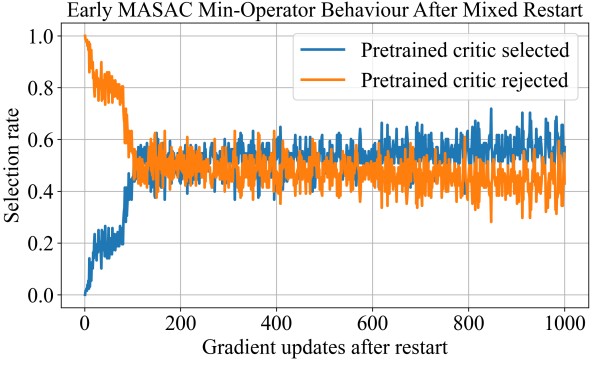

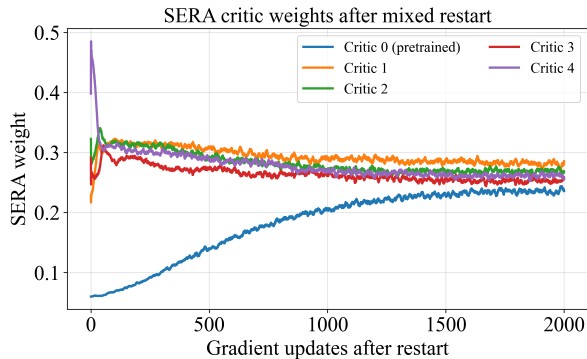

(a) Selection frequency of the pretrained critic under the MASAC minimum operator after a mixed restart.

(b) Evolution of SERA reliability weights after the same mixed restart.

Figure 7: Behavior of the pretrained critic after a mixed restart. One critic is initialized from a pretrained checkpoint, while the remaining critics are initialized randomly. At the beginning of training, the pretrained critic produces value estimates that differ from those of the randomly initialized critics and, consequently, contributes less to the aggregated target. As training proceeds, the disagreement across the ensemble reduces. This behavior is similar in both SERA and MASAC, as both operators could not have any idea of whether the pretrained critic is informative or not.

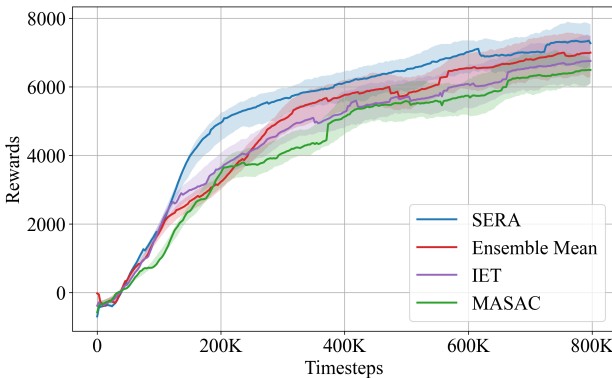

Figure 8: Comparison of different methods with SERA extended upto 800K timestep in 6 agent Halfcheetah

This experiment clarifies the case of an *informative outlier*. During training, the true target value is unavailable, so SERA cannot determine whether an isolated critic is genuinely more accurate or simply producing a noisy estimate. Instead, like other operator-based methods (MASAC and MATD3), it relies only on the information available from the ensemble itself and continuously updates each critic's influence as the critics evolve during training.

It is important to note that SERA does not permanently suppress critics that temporarily disagree with the ensemble. The reliability weights are recomputed at every target update, and the exponential weighting mechanism only reduces the influence of critics exhibiting large deviations from the ensemble median. As training progresses and the critics become more consistent, their corresponding weights increase automatically. Thus, the proposed weighting mechanism performs adaptive attenuation rather than hard rejection.

# F   Extension to 800K timestep

As suggested by the reviewer, we extended the training horizon from 300K to 800K timesteps. Figure 8 presents the reward curves (mean ± standard deviation) over four random seeds. At 800K timesteps, SERA achieved a final reward of $7108 \pm 801$, compared to $6702 \pm 819$ for Ensemble Mean, $6623 \pm 799$ for IET, and

$6431 \pm 831$ for MASAC. SERA also consistently achieves the highest normalized AUC ($5502.32 \pm 391.08$), outperforming Ensemble Mean ($4802.22 \pm 431.91$) IET ($4611.98 \pm 401.22$) and MASAC ($4401.70 \pm 421.78$).

For reference, after **300K** training timesteps, SERA achieved final reward improvements of **14.3%**, **19.8%**, and **22.7%** over Ensemble Mean, IET, and MASAC, respectively. The normalized AUC exhibited a similar pattern, with improvements of **12.04%**, **17.27%**, and **23.56%**, indicating that SERA accumulated higher returns throughout training. When the training horizon was extended to **800K** timesteps, the gains in final reward decreased to **6.1%**, **7.3%**, and **10.5%**, while the normalized AUC improvements remained substantial at **14.6%**, **19.3%**, and **25.0%** over the same baselines. This trend suggests that the competing methods continue to improve with longer training, thereby reducing the gap in final performance. At the same time, the consistently larger improvements in normalized AUC indicate that SERA attains strong performance earlier and maintains this advantage over a considerable portion of the training process. Therefore, although extending the training budget narrows the difference in the final rewards, the learning curves and normalized AUC results show that the sample-efficiency advantage of SERA is retained.

## G   Future Work

Future work will investigate extending the use of SERA's reliability estimates beyond target aggregation. One possible direction is to use these estimates to adapt exploration-related components, such as the entropy-temperature parameter in entropy-regularized reinforcement learning. The effectiveness of these extensions across different actor-critic algorithms will be studied in future work.

