# OpenReview forum: "SERA: Soft Ensemble Reliability Aggregation for Robust Multi-Agent Reinforcement Learning."
_TMLR — Under review for TMLR_

### Review · Reviewer_ABPg · 2026-06-19

**Summary Of Contributions:**

In multi-agent reinforcement learning (MARL), critic learning is unstable due to variance from multiple sources. This paper presents Soft Ensemble Reliability Aggregation (SERA), which constructs temporal difference (TD) targets as a convex combination of ensemble critics, where weights are assigned according to each critic's deviation from the median. Additionally, SERA introduces a variance-adaptive learning rate for each critic, inspired by the Kalman filter, which modulates update strength based on TD error uncertainty and target estimation variance. Together with architectural diversity and separate initializations, these mechanisms aim to decorrelate ensemble members. Under identical and independent critic noise assumptions, SERA is shown to achieve lower variance than any individual critic. Empirically, SERA outperforms MARL baselines on cooperative continuous control tasks and ensemble-based baselines on single-agent tasks.

**Audience:**

Yes

**Audience Explanation:**

The paper presents an interesting idea. Variance reduction in centralized critics is a well-recognized challenge in cooperative MARL, and reliability-aware ensemble aggregation is a reasonable approach to address it. It requires a bit more work to show exactly where SERA would work better than a simple ensemble averaging. Based on that, the authors should select the task that highlights that importance.

**Broader Impact Concerns:**

"No broader impact concerns."

**Claims And Evidence:**

No

**Claims Explanation:**

1. **The theoretical justification for SERA's advantage over ensemble averaging is inconsistent.** In Proposition 4.3 authors show that under iid assumption, SERA has lower variance than any of the individual critics (The proof has a problem but that is deferred to the “requested changes section”). This result is very weak since any convex combination of critics (with at least 2 non-zero weights) would yield this result. Then, they correctly note that under the iid assumption, the arithmetic mean of the critics would yield the lowest variance. But, immediately after that the authors argue that under non-identical critics, SERA would produce a lower variance since it down-weights the outliers. This claim is not clear and does not have any supporting formal discussion. The proposition and the comparison paragraph therefore implicitly operate in different regimes without acknowledging the inconsistency.
2. **The empirical claims are not statistically significant once multiple comparisons are accounted for.** For empirical results, the statistical tests are only shown for the multi-agent setting, with no significance test reported for single-agent setting. Authors test SERA against MASAC (p=0.021), IET (p=0.029) and ensemble mean (p=0.072) baselines, claiming that the comparison against MASAC and IET is statistically significant (assuming that they used the threshold of 0.05). However, no correction for multiple comparisons is applied. After Bonferroni correction, the adjusted threshold becomes (0.05/3 = 0.0167), under which **none** of the comparisons remain statistically significant. Furthermore, the comparison against Ensemble Mean is particularly important since Ensemble Mean shares the variance-adaptive learning rate with SERA, meaning this comparison specifically isolates the paper's primary novel contribution (i.e., the median-anchored soft aggregation). The failure to achieve statistical significance here (p=0.072) suggests that the core proposed mechanism may not provide a reliable benefit over simple ensemble averaging.

**Requested Changes:**

Technical issues:
1. Refer to “Are the claims made in the submission supported by accurate, convincing and clear evidence?” section.
2. Lemma 4.1 is redundant. This follows trivially from the definition. This should at-most be a remark.
3. Proposition 4.3 proof requires $\alpha \in (0,1)$, while the definition has $[0,1]$.
4. Lemma 4.4 proof requires $\epsilon_t$ and $e_t$ to be zero mean. This assumption is not explicitly mentioned. Also, what does zero-mean noise mean in this context? Does the critic already have a good estimate of the true value? TD targets are biased by definition under function approximation. Assuming zero-mean implicitly assumes the target network is already near-optimal, which defeats the purpose of learning?
5. In Section 1, page 2, you say “Together, these components improve target estimation stability, reduce critic correlation, and enable more sample-efficient learning.” Yet, you never talk about sample efficiency in the paper. There is no formal discussion about sample efficiency, and I think this should be removed from the introduction.
6. In Section 4.1, “This median serves as a stable reference point and reduces the effect of extreme critic estimates.” What does stable mean in this context? The median can differ vastly from one update to another.
7. For empirical results, 300K steps might be too less to understand if SERA’s performance persisted.
8. The authors present an MLE derivation to motivate SERA's weighting mechanism. Under Gaussian critic noise, the likelihood-optimal estimator is the inverse-variance weighted combination. However, SERA assigns weights proportional to $exp(-d_i^{(b)}/\tau)$. They say that since the true variance is unknown in practice, they use the distance from median anchor as a proxy. But, the variance and distance measure fundamentally different quantities. These two quantities are structurally different and do not approximate each other in any principled sense. The authors should either provide a formal justification for why exponential deviation weighting approximates inverse-variance weighting in this setting, or be more modest about the MLE connection and present the weighting scheme purely as a heuristic.

Writing issues:

Page 2, “Following (Lyu et al., 2021) “ cite as noun.

Page 4, “For more details, the readers are referred to (Lowe et al., 2017).” cite as noun.

Page 5, “This is used by prior works like (Lee et al., 2021; Liang et al., 2022; Fujimoto et al., 2018).” cite as noun.

---

> ### Author Response · Authors · 2026-07-17
>
> We thank the reviewer for the valuable comments and suggestions. Because of the rebuttal's character limit, our responses below are necessarily brief. The revised manuscript contains the complete explanations and all corresponding revisions.
>
> 1(a). We agree that the original presentation did not clearly separate the homogeneous and heterogeneous critic settings, which could give the impression that SERA was intended to outperform uniform averaging under the standard i.i.d. assumption. This was not our intention. To clarify this, we revised Section~4.1 (Comparison with Mean Aggregation) and added a controlled synthetic study (Appendix B.2). For homogeneous critics, we explicitly state that uniform averaging is the minimum-variance estimator and the synthetic results confirm this behavior. We then discuss the heterogeneous setting separately, which is more representative of practical deep reinforcement learning, where critics naturally differ because of their architectures, initialization, and optimization history. In this setting, the synthetic experiments show that SERA can achieve lower empirical variance than both uniform averaging and a single critic. These revisions clearly separate the two settings and ensure that the conclusions are interpreted in the appropriate context.
>
> 1(b). We strengthened the statistical evaluation by increasing the number of random seeds from 5 to 10, performing environment-wise Welch's t-tests with Bonferroni correction, and extending the analysis to the single-agent benchmarks. With the additional seeds, several comparisons that were previously not significant now satisfy the Bonferroni-adjusted threshold. We also report significance for both the normalized AUC. We agree that Ensemble Mean is the most appropriate baseline because it shares the proposed variance-adaptive critic learning rate. The revised results show that, although the improvement in the final return is not statistically significant against Ensemble Mean in every environment, the normalized AUC remains significantly higher. This indicates that the primary advantage of SERA lies in faster and more stable learning rather than consistently higher asymptotic performance. The complete statistical analysis has been added to Appendix D.2.
>
> 2. Following the reviewer's suggestion, Lemma 4.1 has been revised as Remark 1, since it follows directly from the definition.
>
> 3. We thank the reviewer for identifying this typo in the definition of $\alpha$. The definition of $\alpha$ has been corrected to $\alpha \in (0,1)$, which is consistent with the assumptions used in Proposition 4.3 (now 4.2)
>
> 4. In the revised manuscript, we explicitly state the assumptions $\mathbb{E}[e_t]=0$ and $\mathbb{E}[\varepsilon_t]=0$ in Lemma 4.3 (previously 4.4) and expand Remark 3 to clarify that these assumptions are introduced only to facilitate the theoretical analysis and are not intended to imply that the target network is already accurate or near convergence. We agree that TD targets are biased in practice. The zero mean assumption was used only to derive a variance-optimal adaptive step size in an idealized setting. We also added a discussion showing that the same derivation continues to hold when the critic estimation error and the TD target error share a common constant bias. This discussion has been included in the revised appendix and Remark 4.
>
> 5. We added a normalized area under the learning curve (AUC) analysis, which provides a quantitative measure of sample efficiency throughout training (Appendix D).
>
> 6. We apologize for the confusion. By "stable reference point", we intended to mean robustness against outliers rather than temporal stability across updates.
>
> 7. We extended training on the representative 6-Agent HalfCheetah benchmark from 300K to 800K environment interactions. The results show that SERA maintains the best final performance and the highest normalized AUC. These results have been added to Appendix F of the revised manuscript.
>
> 8. We apologize for the inconsistency. We have therefore revised this section to explicitly state that SERA does *not* approximate the inverse-variance estimator. Instead, MLE provides a theoretical motivation for the proposed weighing scheme. Under a Gaussian critic error model, we show that the expected deviation from the ensemble median increases monotonically with the critic uncertainty, providing the basis for assigning smaller weights to less reliable critics. To support this analysis, we also added a synthetic study (Appendix B.4) that evaluates the relationship between critic uncertainty, deviation from the ensemble median, and the resulting SERA weights across heterogeneous critic ensembles. The results consistently show that critics with higher uncertainty exhibit larger deviations and receive lower weights, supporting the intuition. We therefore renamed the section as "Likelihood motivation and empirical support".
>
> All other writing issues are also revised as per suggestion.

---

### Review · Reviewer_oEZZ · 2026-06-28

**Summary Of Contributions:**

The manuscript proposes SERA, an ensemble-critic method for (primarily) cooperative MARL under CTDE. Two components have been introduced in this work: (1) a reliability-aware target that takes the median of the target critics as a robust anchor, soft-weights the non-median critics by closeness to the median, and (2) a variance-adaptive per-critic learning rate. Experiments on 5 multi-agent (MaMuJoCo + PettingZoo) and 3 single-agent MuJoCo tasks, 10 seeds, report gains "up to 41.1%" (MA) and "up to 31.25%" (SA).

**Additional Comments:**

I have a question about the motivation and practical intuition behind the reliability weighting in SERA. The method appears to treat critics that deviate strongly from the median as less reliable and therefore downweights them. However, in some settings, an outlier critic with high variance may still contain a valid or useful signal, especially if it captures a mode or region of the value landscape that the other critics smooth over. Could the authors clarify how SERA is expected to behave in such cases? In particular, have they tested scenarios where the “outlier” critic is not simply noisy or erroneous, but contains an informative signal that is softened by the rest of the ensemble?

**Audience:**

Yes

**Audience Explanation:**

The manuscript addresses an important problem in a multi-agent setting and will be of interest to the audience of TMLR.

**Claims And Evidence:**

No

**Claims Explanation:**

1. The proof of Proposition 4.3 treats the effective weights $\widetilde{w}_i$ as fixed when computing

$$
\operatorname{Var}(Q^*_{\mathrm{SERA}}) = \sigma^2 \sum_i \widetilde{w}_i^2.
$$
However, the weights are deterministic functions of the realized errors. Specifically, with

$$
Q^*_{\bar{\theta}_i} = y^\star + \varepsilon_i
$$
and the median anchor

$$
Q^*_{\mathrm{med}} = y^\star + \varepsilon_m,
$$
the deviation is
$$
d_i = |\varepsilon_i - \varepsilon_m|,
$$
so

$$
w_i = \operatorname{softmax}(-d_i / \tau)
$$
is a function of ${\varepsilon}$.

Therefore, the step conditioned on the weights,

$$
\operatorname{Var} = \sum_i \widetilde{w}_i^2 \operatorname{Var}(\varepsilon_i),
$$
does not appear to be correct as written. Conditioning on $\widetilde{w}$ changes the law of $\varepsilon_i$. A small $w_i$ is precisely evidence that $|\varepsilon_i - \varepsilon_m|$ is large, so in general,

$$
\operatorname{Var}(\varepsilon_i \mid \widetilde{w}) \neq \sigma^2,
\qquad
\mathbb{E}[\varepsilon_i \mid \widetilde{w}] \neq 0,
$$
and the cross-terms

$$
\mathbb{E}[\varepsilon_i \varepsilon_j \mid \widetilde{w}]
$$

need not vanish. Hence, the conditional independence used in the proof no longer holds once the weights depend on the same realized errors being aggregated. Based on this, I am wondering if the authors can clarify this point and, if valid, adjust the proof accordingly.

2. In the contribution segment, the author reports the work has been evaluated on "nine benchmarks" for the MA setting, but I could only find five. Please clarify this inconsistency in the manuscript.

3. The entropy mechanism used in Algorithm 1 has been briefly reported in the main body, and my understanding of it is that it serves as a major part in the method. Therefore, the authors should ablate this component.

**Requested Changes:**

1. The proof for Proposition 4.3 should explicitly account for the dependence between the weights and the critic errors. The authors should either provide a corrected argument under clearly stated assumptions, or weaken the theoretical claim and support it with an empirical variance-reduction study on a controlled synthetic critic ensemble.

2. The authors should clarify whether four benchmarks are missing from the manuscript, or whether the “nine benchmarks” statement is an error. This should be corrected consistently across the abstract, contributions, experimental section, tables, and figure captions.

3. Algorithm 1 introduces an entropy-related adaptive temperature mechanism, and my understanding is that this component is not merely an implementation detail but a potentially important part of the method. Since it appears to affect the TD target and may contribute materially to the reported performance, the authors should include an ablation that isolates its effect. For example, they should compare full SERA against SERA without the adaptive entropy mechanism, while keeping the ensemble aggregation and adaptive learning-rate components unchanged. If this mechanism is not active in the reported experiments, the manuscript should state that clearly and avoid attributing performance gains to it.

---

> ### Author Response · Authors · 2026-07-17
>
> 1) We sincerely thank the reviewer for identifying this issue in the original Proposition~4.3 (now its 4.2) and apologize for the error in the previously submitted proof. We agree that the previous analysis incorrectly treated the adaptive SERA weights as fixed and  independent of the errors, although the weights are computed from the same critic estimates. We have therefore revised the original proposition with a sufficient condition that explicitly accounts for this dependence through the covariance term $\sum_{i=1}^{N}\operatorname{Cov}(\tilde{w}_i,\varepsilon_i^2)<0$. Unlike the previous proposition, the revised result does not rely on an independence assumption and instead shows that SERA reduces the variance relative to a single critic whenever this covariance is negative.
>
> Proving this condition analytically is difficult as the SERA weights depend on the relative deviations $(e_i-e_m)$. We therefore followed the reviewer's suggestion and evaluated it empirically using a controlled synthetic ensemble with a known target. The weights are computed exactly as in SERA, without assuming $Q_{\mathrm{med}}\approx y^\star$, and the covariance is measured directly over $10^6$ randomly generated samples covering multiple softmax temperatures, random seeds, ensemble sizes, and error distributions. In every setting, the covariance remains negative, providing empirical support for the revised sufficient condition. The complete proof and empirical validation are included in the Appendix.
>
> 2) We thank the reviewer for identifying this typographical error. The reference to "nine benchmarks" was a typographical error. We have corrected this throughout the manuscript to consistently reflect that the proposed method is evaluated on five benchmark environments.
>
> 3) We thank the reviewer for this valuable observation and apologize for the lack of clarity in the original manuscript. The adaptive entropy mechanism included in the appendix of the previous version was not part of the SERA framework evaluated in this work. It was included only as a potential extension for entropy-regularized algorithms such as SAC. Since the other algorithms considered in our experiments, namely MATD3, MAPPO, and MADDPG, do not use an adaptive entropy-temperature mechanism, this extension is not applicable to them. For this reason, it was not presented as part of the main method and was not used in any of the reported experiments or ablation studies. Accordingly, the performance improvements reported in the paper are unrelated to this optional extension. To avoid any ambiguity, we have removed Algorithm 1 and the associated discussion from the appendix in the revised manuscript. We now refer to it only briefly as a possible direction for future work.
>
> **Additional Comment**: We thank the reviewer for this thoughtful comment. Since the true target is unavailable during training, no aggregation method can know whether a critic that disagrees with the ensemble is more accurate or simply less reliable.
>
> To examine this case, we conducted an additional experiment where one critic was initialized using the parameters of a model trained for 200K timesteps, while the remaining critics were initialized randomly. This creates a controlled setting in which one critic begins with prior knowledge and therefore produces value estimates that differ from the rest of the ensemble, making it possible to study the behavior of SERA when the outlier may carry useful information rather than being purely noisy. We conducted the same experiment for MASAC but with two critics.
>
> For MASAC, immediately after the restart, the pretrained critic produces value estimates that differ from the randomly initialized critic. Under the standard MASAC minimum operator, it is selected less frequently at first because its estimates are generally higher. As the other critics learn and their predictions become more similar, it is selected more often. This shows that the minimum operator cannot determine whether a disagreeing critic is genuinely more informative or simply less accurate because of estimation errors.
>
> SERA faces the same uncertainty because it has no access to the true target during training. Rather than making a hard decision about whether the pretrained critic is correct or incorrect, it adjusts the contribution of each critic continuously. Since the pretrained critic initially disagrees with the rest of the ensemble, it is assigned a smaller reliability weight, but it is never excluded from the aggregation. As training progresses and the remaining critics become more consistent with its predictions, its reliability weight increases automatically.
>
> More broadly, this limitation is shared by all existing aggregation strategies. Without access to the true target, no existing aggregation strategy can reliably distinguish informative disagreement from estimation error during training. We have added this experiment and the corresponding discussion to Appendix Section~E.

---

### Review · Reviewer_gQFh · 2026-07-21

**Summary Of Contributions:**

**Contributions**
The work makes two major technical contributions and two further empirical contributions.

For the technical contributions, the work proposes
1. a novel ensemble value aggregation technique to compute temporal difference (TD) targets in (multi-agent) reinforcement learning to stabilize training.
   In short, this technique uses an ensemble of value functions to compute the median value estimate as an anchor and then computes weights for each value estimate based on its distance to the median value with estimates closer to the median value receiving a larger weight (considered more reliable). Target values are then computed as a weighted sum of all value estimates.
2. an adaptive learning rate scheduling approach in which the learning rate of each critic within the ensemble is varied based on an exponential moving average (EMA) of the variance of TD errors of the critic, with higher variance leading to a reduction in learning rate as the computed gradients might be considered less robust.

For the empirical contributions, the work demonstrates through empirical experiments that the proposed SERA approach
1. improves evaluation performance, measured by cumulative rewards, across five multi-agent continuous control tasks compared to standard baselines and similar ensemble-based approaches.
2. improves evaluation performance, measured by cumulative rewards, across three single-agent continuous control tasks compared to ensemble-based approaches.

**Strengths**
Training instability during TD learning is a well known challenge in reinforcement learning, that extends to and is further expanded in multi-agent settings. Therefore, I believe that the problem tackled in this work is highly relevant to the RL literature and the proposed SERA aggregation and learning rate adaptation schema are interesting contributions.

I also appreciate that the work combines theoretical analysis to motivate the proposed method, albeit the theoretical analysis feels slightly disconnected from the proposed method and empirical results. Lastly, I appreciate that the appendix provides more details on supplementary experiments but it reads slightly like a laundry list of ideas. Further contextualization or integration with the main paper claims and experiments would benefit the overall work, as related to Weakness 1 listed below.

**Weaknesses**
1. In my opinion, the classification of this work as regular submission with less than 12 pages of main content is questionable given the work defers necessary evidence and details to support core contributions to the 20 pages of appendix. Below are some parts of the appendix that I'd expect to see in the main paper (see also further related comments in discussion of "Claims and Evidence"):
	1. Section 4 defines the target value computation and adaptive learning rate schemes as the novel contributions referred to as SERA. However, they do not define a full single-agent or multi-agent RL algorithm and without pseudocode of Algorithm 1 in the appendix, it is unclear what algorithm is being evaluated as "SERA" in Appendix 5. The main paper merely refers to this pseudocode, and states that MASAC was chosen as the base updating rules. For example, without reading Algorithm 1 in the appendix or adding a more detailed description of the complete algorithm, it is entirely unclear whether there are separate trained policy networks, or how the ensemble of value functions is used to train these policies. This is only clarified in Algorithm 1 and the initial two paragraphs of appendix C.
	2. Section 4.2.2 introduces an adaptive learning rate scheduling but appendix C.1.4 states that the evaluated SERA algorithm in fact uses a variation of the introduced schema that uses additional hyperparameters to scale a base learning rate. I find it questionable that this detail is only mentioned in the appendix without any reference in the main paper.
	3. I consider it reasonable practise to defer to the appendix to state hyperparameter details for reproduction but given it is central to the proposed SERA algorithm, I find it surprising that, as far as I can tell, the main paper never specifies how many value functions are used within the SERA ensemble $K$.
	4. I would expect at least a brief discussion of future work in the main paper but this is deferred to the appendix G.
2. The depiction of related literature and their contributions in this work is sometimes misleading or inaccurate. Below are some examples that should likely be addressed:
	1. SUNRISE is stated to "leverage ensemble diversity to enhance exploration and robustness, rather than directly addressing variability in value estimation during critic updates" but the SUNRISE algorithm explicitly proposes value estimation weighting based on the reliability of value estimates so I'd argue it explicitly addresses variability in value estimation for critic updates.
	2. EMAX and IET are stated to "leverage ensembles primarily to encourage exploration, representation diversity, or robustness, rather than to explicitly control instability arising from noisy value estimates during critic learning. As a result, estimation variance remains largely untreated as a first-class factor in stabilizing critic updates." However, EMAX explicitly proposes target computation aggregated across the ensemble to "reduce the variability of gradients and improve the stability of training" (quote from EMAX paper).
	3. Bootstrapped DQN is mentioned in a list of methods that aim to address overestimation, but the work does not try to address overestimation. In fact, value overestimation is never mentioned in the Bootstrapped DQN work.
	4. I would argue that a fundamental reference in related literature is missing in van Hasselt, Hado. "Double Q-learning." _Advances in neural information processing systems_ 23 (2010). To the best of my knowledge, it was one of the first to explicitly discuss the overestimation as a result of the Jensen inequality in the maximization step and to propose a technique to address overestimation in Q-values. It predates deep approaches such as Double DQN or the work by Thrun & Schwartz cited in this context.
3. The work lacks clarity in several cases of notation / equations, statements, and empirical details:
	1. Undefined terms or lack of clarity in equations
		1. Equation (7): $Q_t$ is undefined.
		2. Equation (10): how is $y_j$ defined here? Is it the SERA target or another target value?
		3. Equation (11): $P_t^{(i)}$ appears to be computed as an EMA of an EMA of the variance of critic i's TD error. Why do you compute a nested EMA rather than a single EMA?
		4. Equation (13): $S$ appears undefined. I presume it is the minibatch size?
		5. Equation (13): is this $\alpha_Q$ the same as in Equation (10)? Why does that EMA factor appear here?
	2. SERA introduces several hyperparameters, with the impact of several of them on the algorithm performance not being analyzed. Furthermore, $\alpha$, $\alpha_Q$, and $\alpha_P$ are introduced but the work does not report their chosen values within experiments.
	3. The Ensemble Mean baseline is never defined. How does it differ from MASAC and SERA?
	4. Subsection 4.2: "Second, the critics are designed with heterogeneous hidden-layer architectures [...]" - until reading appendix C, it was not clear to me that "heterogeneous hidden-layer architectures" refers to each critic within the ensemble using a different network architecture.
	5. How were the hyperparameters of all algorithms in SERA and baselines tuned? This question also applied to the ablations presented in Figure 4b - did you tune their hyperparameters or just take the hyperparameters from the tuned SERA algorithm?
	6. Figure 1 lacks any details about the presented data. Which tasks are these results computed in? Which algorithms are used during learning? If the variance in multi-agent is a result of multiple agents, can you show that it increases with the number of agents increasing in a comparable task?
4. The depiction of SERA as a multi-agent algorithm first, and a single-agent algorithm second, is confusing to me. As far as I can tell, no component of the SERA algorithm is specific to multi-agent settings and they could equally be applied to single-agent RL algorithms. I understand that the work motivates that the multi-agent setting further exacerbates target value instability and variance but I struggle to see SERA as a multi-agent reinforcement learning algorithm. I consider it a strength of SERA that its techniques can also be valuable in multi-agent settings but, as presented, I find it difficult to delineate these results and I believe they would benefit from clear context and motivation.
5. Lemma 4.3 assumes that the critic estimation and target errors are uncorrelated but this appears unreasonable given target values are computed with target networks that are time-delayed versions of the main critic networks and, thus, inherently dependent on each other.

**Additional Comments:**

Below, I provide a list of further minor comments that should be addressed but are not critical to this work.
- The reference to the EMAX work appears messed up. It is missing the venue, year, the list of authors is missing commas, and the author ordering appears wrong based on a quick search.
- The MADDPG citation for Rowe et al. 2017 states "OpenAI" in the middle of the author list.
- The introduction mentions that "non-zero learning rates" can result in unstable value estimates but learning rates should always be non-zero.
- Equation (1) has a typo in the notation of the target network parameters.
- Section title "Related Works" is written in all capitalisation.

**Audience:**

Yes

**Audience Explanation:**

Novel approaches to obtain more robust target values for temporal difference learning are of high relevance to the reinforcement learning audience within TMLR.

**Broader Impact Concerns:**

I have no concerns about ethical implications of this work that require further discussion.

**Claims And Evidence:**

No

**Claims Explanation:**

The work makes several claims that are insufficiently supported with evidence. I would expect all of these to be addressed by providing convincing evidence and/ or adjusting made claims accordingly.

Below are claims related to core contributions of the work that I believe are insufficiently supported with evidence.
1. In subsection 5.1, the work states that "The results in Fig. 2 show that SERA consistently achieves stronger performance than all compared base lines across the tested environments" but from reported p values in subsection 5.3, it is unclear whether performance gains over the Ensemble Mean baseline can be considered statistically significant/ meaningful.
2. The key claim about the target aggregation of SERA is that it improves the stability/ reduces the variance of target values but the work provides insufficient evidence for this.
	1. Figure 4c shows that SERA's targets exhibit a lower variance than the base MASAC algorithm but no such comparison is provided for simple average or median aggregation across the ensemble which has been used by prior work. Without such comparisons, the work provides no evidence that the novel weighted aggregation of SERA provides any gains over previously proposed alternatives. Also, I would expect to see shading as variation of these variances values across multiple random seeds to judge whether differences are likely due to random variations or not.
	2. Similarly, Figure 4a shows the bias of initial state for several MARL baselines, but no comparison is provided to the Ensemble Mean or IET ensemble baselines. Does the SERA target computation lead to more accurate value estimates than these existing ensemble methods at identical ensemble size and tuning?
	3. The analysis of the target aggregation also appears rather shallow. The work claims that uniform value aggregation is insufficient so I would expect some form of analysis as provided in Figure 7b in Appendix E for some of the main results. How do SERA weights vary throughout learning and critics within the ensemble? Seeing notable differences between critic weights would be necessary to support the claim that the SERA aggregation is adding value over uniform aggregation techniques.
3. The key claim about the adaptive learning rate scheduling is that it helps to learn and retain distinct critics within the ensemble.
	1. The work currently provides no intuition as to why adjusting the learning rate based on trustworthiness of the prediction would specifically help with decorrelation of critics within the ensemble. It motivates that this might be a good idea to minimize the critic error but the connection to decorrelation is missing to me.
	2. The only concrete evidence to this claim I can see is Figure 6 in Appendix B.5. I would expect this evidence towards a central claim of a contribution to be within the main paper. Also, the figure and appendix section lack context and details.
		1. What is the task for which the correlation is computed?
		2. The figure compares "Adaptive LR" and "No Adaptive LR". Do these correspond to the SERA algorithm and the SERA-NoLR ablation, which I'd expect to represent SERA with just a fixed learning rate across all critics?
		3. The correlation is just shown at 50k timesteps. How does this change throughout learning and can you show the correlation next to learning curves, since no learning would naturally lead to decorrelation for any learning rate schema.
	3. Similar to the target aggregation, I find the provided analysis rather shallow. I would expect to see some analysis and/ or visualization that shows the learning rate variation throughout training for different critics.

Below are some further claims made throughout the paper that I consider to be insufficiently supported with evidence:

4. Subsection 5.2: "As the multi-agent study already includes widely used CTDE-based methods such as MATD3, MASAC, and MAPPO, their corresponding baselines across three single-agent MuJoCo environments."
   I do not find this statement convincing. MARL and single-agent RL tasks vary significantly in tasks and learning dynamics so I would still expect to see comparison to at least one single-agent baseline. Given the SAC algorithm appears to be the foundation for SUNRISE and SERA, I would suggest to add comparison to SAC to provide sufficient grounding of results.
5. Subsection 5.3: "Even with this additional cost, the overhead remains moderate relative to the gains in training stability and learning efficiency."
   Depending on the cost of obtaining samples within a task, it might be significantly cheaper to train MASAC/ Ensemble Mean for slightly longer with the same total compute/ time cost to obtain potentially higher performance than SERA.

**Requested Changes:**

**Critical changes**
1. Include a definition of the full SERA algorithm as evaluated in Sections 5.1 and 5.2 in the main paper. (See weakness 1)
2. Provide clear evidence for the claim that the SERA target aggregation leads to more stable target values than standard mean/ median aggregation across the ensemble, and show analysis of the weighting scheme throughout training. (See claims & evidence 2.)
3. Provide clear evidence for the claim that the SERA adaptive learning rate leads to larger variation of critics within the ensemble than using a fixed learning rate, and show how the learning rate varies throughout training. (See claims & evidence 3.)
4. Define the Ensemble Mean baseline and clarify how it differs from MASAC and SERA. (See weakness 3.3)
5. Provide comparison to the SAC base algorithm in the single-agent experiments. (See claims & evidence 4.)
6. Correct misleading or inaccurate statements about related work. (See weakness 2)
7. Define and clarify unclear notation in Equations. (See weakness 3.1)
8. Define the chosen values for all hyperparameters of SERA in the Appendix ($\alpha$, $\alpha_Q$, and $\alpha_P$ are not specified). (See weakness 3.2)
9. Specify how hyperparameters of baselines and SERA have been tuned/ chosen for each experiment, including the ablations (See weakness 3.5)


**Changes that would strengthen the work**

10. Provide more contextualization as to whether SERA should be seen as a multi-agent reinforcement learning algorithm (as suggested by the paper title and focus on MARL in the evaluation), or a single-agent reinforcement learning algorithm (as suggested by the algorithm having no multi-agent-specific component). (See weakness 4)
11. As presented, the empirical evidence of the work is limited to few similar continuous control tasks which I would consider the bare minimum to substantiate any claims about the resulting performance. Experiments beyond five multi-agent and three single-agent mujoco tasks would significantly strengthen any claims about performance gains resulting from SERA, in particular if additional tasks would focus on environments where stable value estimation is challenging, e.g. due to high degrees of stochasticity or partial observability just to name two examples. That being said, if claims are nuanced and well substantiated, I do not believe that evaluation in more tasks is necessary to justify publication.
12. Figure 1 would benefit from further details to contextualize and motivate the core problem SERA tries to tackle. (See Weakness 3.6)